

# Security durability assessment through fuzzy analytic hierarchy process

Alka Agrawal[1], Mohammad Zarour[2], Mamdouh Alenezi[2],
Rajeev Kumar[1] and Raees Ahmad Khan[1]

[1] Information Technology, Babasaheb Bhimrao Ambedkar University, Lucknow, Uttar Pradesh, India
[2] College of Computer & Information Sciences, Prince Sultan University, Riyadh, Saudi Arabia

## ABSTRACT

**Background:** Security is an integral aspect of the development of quality software. Furthermore, security durability is even more imperative and in persistent demand due to high investment in recent years. To achieve the desired target of efficacious and viable durability of security services, there needs to be nodal focus on durability along with security. Unfortunately, the highly secure design of software becomes worthless because the durability of security services is not as it should be.

**Methods:** Security durability attributes have their own impact while integrating security with durability and assessment of security durability plays a crucial role during software development. Within this context, this paper estimates the security durability of the two alternatives versions of a locally developed software called version 1 and version 2. To assess the security durability, authors are using the hybrid fuzzy analytic hierarchy process decision analysis approach.

**Results:** The impact of the security durability on other attributes has been evaluated quantitatively. The result obtained contains the assessment of security durability. The study posits conclusions which are based on this result and are useful for practitioners to assess and improve the security life span of software services.

# INTRODUCTION

Security specialists are confronting with various issues to comprehend the new security challenges at the initial phases of software development. There is a ceaseless burden on the developers to maximize the development and at the same time lessen the expense and time invested in security to optimize the financial dividends of the organization. The nature of software development is becoming even more perplexing at each step with the requirement for security expanding in each field. Evaluating and looking after confidentiality, integrity, and availability (CIA) amid phases of programming advancement has ended up being an extraordinary task as compared to other approaches to get more secure software (*Tekinerdogan, Sozer & Aksit, 2008*; *Subashini & Kavitha, 2011*). Security in the product must be consolidated in software development advancement from the earliest starting point and it ought to proceed till the software is being used (*Boegh, 2008*). Consolidating security amid security improvement prompts reduction of development budget and effort. It must not be forgotten by security specialists when the

Corresponding author
Raees Ahmad Khan,
khanraees@yahoo.com

advancement of software security development is finished or it ought not to be dealt with at the late stage of software development.

As per the predictions done by 31 experts of software security of PhoenixNAP IT services at the end of year 2018, machine learning technologies with smartphones are going to be new challenges to conquer by security practitioners (*PhoenixNAP Global IT Services, 2018*). These predictions produced major contribution in the area of life span of security of software including many macro levels direct or indirect findings. The estimation practice at early stage is beneficial for secure and durable software development. Also, according to a technical report, Software-as-a-Service (SaaS) operations and Management Company, about 73% of the organizations expect to shift nearly all of their applications to SaaS by 2020 and want to improve the life-span of services (*Lambert, 2018*). Veracode tested a scan of 400,000 numbers on their clients' software in a 1-year period which started in April 2016 (*Eng, 2018*). In these scans, they found 12.8 million flaws. According to the report, it was found that stakeholders who use antivirus software to scan the improvements of security were able to detect at least one vulnerability during the initial scan. About one in eight were found to be of high or very high severity vulnerability related to life span of security services.

In 2016, companies closed only 58% of vulnerabilities in the same calendar year in which they were found. The percentage of companies that successfully passed checks for weaknesses on the OWASP Top 10 list declined to 35% for internally developed software, down from 39% in the last year's report. Third-party code, which typically has more vulnerabilities, also performed worse year after year as only 23% passed the OWASP Top 10 check. This was down from 25% in the previous year. Globally, the data shows that organizations are trying hard to stay away from vulnerabilities and doing the security checks on a regular basis. Yet there is something missing, and secure software for a long time seemed to be a mirage. Therefore, developers need to understand how to relate security attributes with those of durability and measure the impact of these attributes for enhancing secure life span of software. Assessment of security durability attributes is necessary to ensure long term security (*Lambert, 2018*). Outcomes of evaluation process may allow decision makers to make appropriate decision as well as propel action (*Bishop, 2017*; *Eng, 2018*). However, to be able to take appropriate action, decision makers are not only need to know about security and durability attributes but their mapping also.

Multi-criteria decision analysis (MCDA) approach is a discipline which aims to support experts when they are faced with various conflicting items for evaluation (*Gray et al., 2015*). The MCDA approach is very suitable to take two or more conflicting problems side by side. Various MCDA methods are available including analytic hierarchy process (AHP), fuzzy analytic hierarchy process (Fuzzy AHP), and multi-attribute utility theory (*Dalton, Kannan & Kozyrakis, 2007*; *Gray et al., 2015*). All these approaches are differentiated by the way the objectives and alternative weights that are determined through it. Although AHP is considered good while analyzing a decision in a group, various researchers have found that hybrid AHP is better in providing crisp decisions with their weights too (*Mikhailov, 2003*).

Hence, in order to deal with the uncertainty and ambiguity of human judgment, the authors have used a hybrid version of AHP (also known as Fuzzy AHP) which incorporates fuzzy set theory with AHP methodology (*Mikhailov, 2003*; *Hahn, Seaman & Bikel, 2012*) to evaluate security durability of software services. This paper presents an approach for evaluating life span of security services.

The results help to formulate development strategies to achieve the desired security durability. This may help software developers to come up with durable as well as secure software. According to the structure of the paper, firstly, authors reviewed the literature available on the signified area. In the "Materials and Methods" section, the authors have introduced security durability and are using one of the most famous MCDA techniques which is called the Fuzzy AHP to evaluate weights of the security durability attributes. In the next section of paper with the help of these weights, the authors have categorized the most important attributes at each level and proposed some suggestions to improve the life span of security of software. To evaluate the ratings of the attributes of security durability, two successive versions of a case study have been taken, that is, entrance examination software for Babasaheb Bhimrao Ambedkar University, Lucknow, India (BBAU software). Thereafter, in the next sections, the authors have assessed "Security Durability" and given suggestions for practitioners based on it. In the last section, "Results, Discussion, and Conclusion" have been profiled.

## Literature review

The digital age has made software an elemental aspect of everyone's life in various forms such as to share data, to communicate, to maintain databases, etc. Almost every facet of life today is connected with some kind of software, be it through banking, health, education, engineering, social realms, or others. Hence, all information related to software must be secure and the demand for secure software has increased today. Software security can be termed as the idea to secure software from malicious attacks and fraudulent persons or hackers (*FCW Workshop, 2016*; *Mougouei, 2017*). Many experts have discussed many areas of security including security attributes, security management, security maintenance, etc., but still, there is something missing. Organizations are investing both money and resources to optimize the maintenance of security for improving the life span of the software (*Mougouei, 2017*). Yet, they have not been successful. Some of the pertinent efforts of the practitioners to assess and improve the security of software are discussed below:

*Weir et al. (2019)* proposed a common framework of security assurance for developers. The framework defined the problem in security awareness and organized a 3 months light weighted security assurance workshop. The workshop focused on security assurance. Based on the report, the authors have given a common guideline for developers to improve the skills to increase the security services while the software is in use. The adoption of this process plays a key role in improving software security for the end users. *Dayanandan & Kalimuthu (2018)* evaluated the quality for security analysis. Authors of the paper proposed a framework for quality assessment at software architecture level. The assessment focused on security because it is the key attribute of quality. The relationship

between quality and object oriented design properties has been well established. And Fuzzy AHP method has been used to evaluate the results. For the validation of the framework, authors have used four versions of apache Tomcat series.

*Mougouei (2017)* defined the modeling process through quantitative assessment. The author defined the problem in existing prioritization techniques for security attributes and the needs of prioritization of attributes. These are usually ignored and thus give birth to new but insecure software. To address it, the author proposed to consider the partial satisfaction of security needs when tolerated rather than ignoring those security needs for the future. As a result, this research has contributed a framework that prioritizes and selects security requirements. *Praus, Kastner & Palensky (2016)* examined the security and software architecture through a critical survey. The authors presented a research on software security requirements in building automation. Their paper provided an extensive survey of the security requirements for distributed control applications and analyzed software protection methods. Architecture on the same problem has been defined that works to secure software that runs on different devices or classes. This architecture also prevents attacks on smart homes and buildings.

Along with fixing security issues, the design of security should also be strong. Hence, to improve the security, designing is the main point during secure software development. With the emergence of new threats, new security issues are being generated day by day. Fixing these latest security issues requires more investment in maintenance cost. Time incorporated in security development also increases. However, there is persistent pressure from the users' end to minimize on both the time and cost. Many practitioners are trying to fill the hole of security design so that new threats are removed and security services are enhanced with it. To improve the software's service life, security life span should be improved.

The following literature review underlines the security durability of software services:

*Chen et al. (2017)* defined the maintainability as a big concern for non-durable software. The author described: "Why is it important to measure maintainability and what are the best ways to do it?" Her study discussed that the durability of software is improved by reducing the cost and time involved in maintenance. The author discussed that there are metrics that can help software developers to measure and analyze the maintainability of a project objectively. This research paper addressed the importance of understanding software maintainability, gave a framework and some of the best ways to measure maintainability. In the same year, *Alarifi, Alsaleh & Alomar (2017)* proposed a structured inspection model for thoroughly evaluating the usability and security of internal and external e-banking assets. The authors have also demonstrated the insufficiency of existing security–usability models and have also applied their proposed framework to evaluate five major banks. The results clearly reflect several shortcomings regarding the security and privacy features in banks.

*Kelty & Erickson (2015)* addressed maintainability issues. The authors stated that the design is responsible for less durable software. The authors discussed about achieving durable software with optimal maintenance. According to the authors, the durability of software depends on its different applications such as a social, economic, and cultural field.

Durability is a result of robustness and maintainability. The paper explains maintainability as a never-ending process and hence reduces durability. The authors further suggest finding the ways for ensuring the durability of software by design because it still needs to improve for better user experience. In 2014, Security Standards Council addressed the optimal maintenance process of vulnerability for improving security life span (*Security Awareness Program Special Interest Group PCI Security Standards Council, 2014*). The Council published a special report on the workshop on software measures and metrics to reduce security vulnerabilities. The goal of the report was to gather ideas on how the federal government can identify, improve, package, deliver, or boost the use of software measures, metrics to significantly reduce vulnerabilities and enhance the working life of software with optimal maintainability. The report contains observations and recommendations from the workshop's participants. The report includes position statements submitted to the workshop, presentations at the workshop and related material.

*Ensmenger (2014)* defined that maintainability plays a key role in decreasing the durability of software but the solution to this problem is not given. The author says that software durability and software serviceability are two faces of the same coin. There is a significant issue of long-time services and increased cost spent on the maintenance of software. Further, the author discusses working life of durability which decreases as the time passes. Hence, for long-term software, durability does play a key role. The study also related durability with maintenance, as time wasted upon the maintenance can be reduced considering the factor of durability in s/w. In the end, the author concludes that maintenance can be a central issue in the history of software, the history of computing, and the history of technology if it does not deliver durable software. In this context, software developers should focus on security and durability simultaneously during software development to improve the life span of security as well as software. Further, *Parker (1992)* said that long security life span is needed to improve the user's satisfaction related to protecting user's data. He also discussed the challenges of high maintenance of security during the use of software services. Due to the high maintenance cost of security, practitioners are focusing on security design during a specified life span of software.

According to Nathan Ensmenger, in the early 1960s, the development of the IBM OS/360 operating system has taken 4 years of maintenance time that absorbed more than 5,000 staff years of effort and cost the company more than half-a-billion dollars. This makes it the single biggest expenditure in IBM history (*Ensmenger, 2014*). To solve these types of issues, there is a need to address the security durability during software development. Quantitative assessment is one of the most important methods to address, assess, and solve any issue. Security design during software development is a very crucial task. There are so many factors that affect the security and durability simultaneously including CIA. Every organization has its own methods and logic to develop the security as well as software design. All in all, this is a multiple decision analysis problem in perspective of the durability of security, that's why researchers have taken an MCDA technique to assess the security durability.

### Security durability of software

The importance of software in our lives is growing daily. People's personal and professional lives can greatly be enhanced by the presence of highly secure and durable software and can greatly be imposed upon by the presence of poor quality software. Most complex software systems, such as airplane flight control or nuclear power plants, depend critically upon the durability of their secure software. In today's world, organizations are busy in understanding and mitigating security challenges during the software development life cycle. There are some key characteristics of the security and focusing on those may help to address these challenges directly or indirectly. One of these characteristics is durability. It may also be called as working life or longevity of security (*Ensmenger, 2014*). The security durability of software is highly essential in sensitive fields including the banking sectors, etc. (*Cusick, 2013*). Security is directly involved in the service life of the software. Durability is further directly or indirectly involved in the security of software and vice-versa (*Kelty & Erickson, 2015*). Through the literature review of previous work and best practices, the authors have defined the security durability/ durable security as:

> The ability of software to secure itself for the expected life-span
> or
> The ability of software to withstand attacks for the expected life-span

Durability means how long a software security solution will function effectively and meet the security requirements. There are several reasons for organizations to integrate durable security during software development as:

- To provide longer security in the given service environment, thereby mitigating security challenges (*Boegh, 2008*).
- To reduce maintenance time by reducing the effort needed to fix bugs by delivering durable and secure software (*Bishop, 2017*).

These are two main reasons to examine the security and durability simultaneously for addressing, assessing, and improving the security durability. There are so many attributes of security and durability which are related to each other. These attributes are useful in assessing security durability. Further, the authors' previous works are identified and classify the security durability attributes (*Kumar, Khan & Khan, 2015*) which are discussed in next sections.

## MATERIALS AND METHODS

### Methodology of assessment

Security is one of the most important quality properties of software which is concerned with both end users and developers (*Lambert, 2018*). Security estimation plays a key role in improving the quality of software. Durability plays a key role in enhancing the security life span (*Alarifi, Alsaleh & Alomar, 2017*). To improve the security life span of software,

security durability assessment is essential which may be helpful for security policy, goals, etc., and user's satisfaction. Security cannot be durable until security durability is not measured. To assess security durability, MCDA method is well suited because of the advantage of assessing any attribute with multiple sub properties by this method. AHP is very popular in troubleshooting such problems.

Major benefit of AHP is its relative simplicity with which it handles multiple criteria. AHP allows decision makers to mold a complex problem in a hierarchical structure that consists of the goal, aims, sub-objectives, and alternatives. Traditional methods of AHP cannot be used when there is uncertainty in data (*Security Awareness Program Special Interest Group PCI Security Standards Council, 2014*). To address such uncertainties, fuzzy set theory was first introduced. Many times, priority assessment of different attributes usually fails because of the connection of multiple qualitative criteria. Fuzzy AHP is a suitable evaluation technique capable of handling this kind of problem with uncertain inputs.

### Implementation

In order to address the fundamental difficulty of security durability assessment, researchers have taken a hybrid method, that is, Fuzzy AHP methodology. Although, AHP is considered good while analyzing a decision in a group, various researchers have found that hybrid AHP is better for providing crisp decisions with their weights too (*Goli, 2013*). Hence, in order to deal with the uncertainty and ambiguity of researchers and academicians, the authors have used a hybrid version of AHP (also known as Fuzzy AHP) which incorporates fuzzy set theory with AHP methodology (*Chong, Lee & Ling, 2014*), to evaluate security durability of software. The adopted methodology is given in Fig. 1 that is in the form of a flow chart. The flow chart describes the process of security durability assessment. It has been divided into five phases/steps including planning; fuzzification; fuzzy operations; defuzzification; and analysis, confirmation, and estimation. Planning phase deals with problem recognition, selecting the alternatives for the problem and defines the scope and boundaries of the AHP. Fuzzification phase deals with the preliminary process of methodology including defining the membership function with a scale. Fuzzy operations phase deals with the performance of pair-wise comparison matrixes through triangular fuzzy numbers (TFNs) with the help of the expert's opinions. Defuzzification phase deals with the transformation of fuzzified weights into defuzzified linguistic values while the last phase deals with weights, ratings, and assessment. Further, the last phase also deals with improvement (performance), sensitivity analysis, and validation of the results through statistical analysis. The phase-wise description of the methodology is given in subsections as:

### Planning phase

The problem of security durability is recognized, addressed in previous sections and related attributes of security durability are identified, categorized in previous work of the authors (*Goli, 2013*). AHP is used as a decision-making tool for estimating the priority numbers for different alternatives with a hierarchical structure of multiple criteria (*Kumar, Khan & Khan, 2015*). According to this research, AHP is best suited for choosing the apt

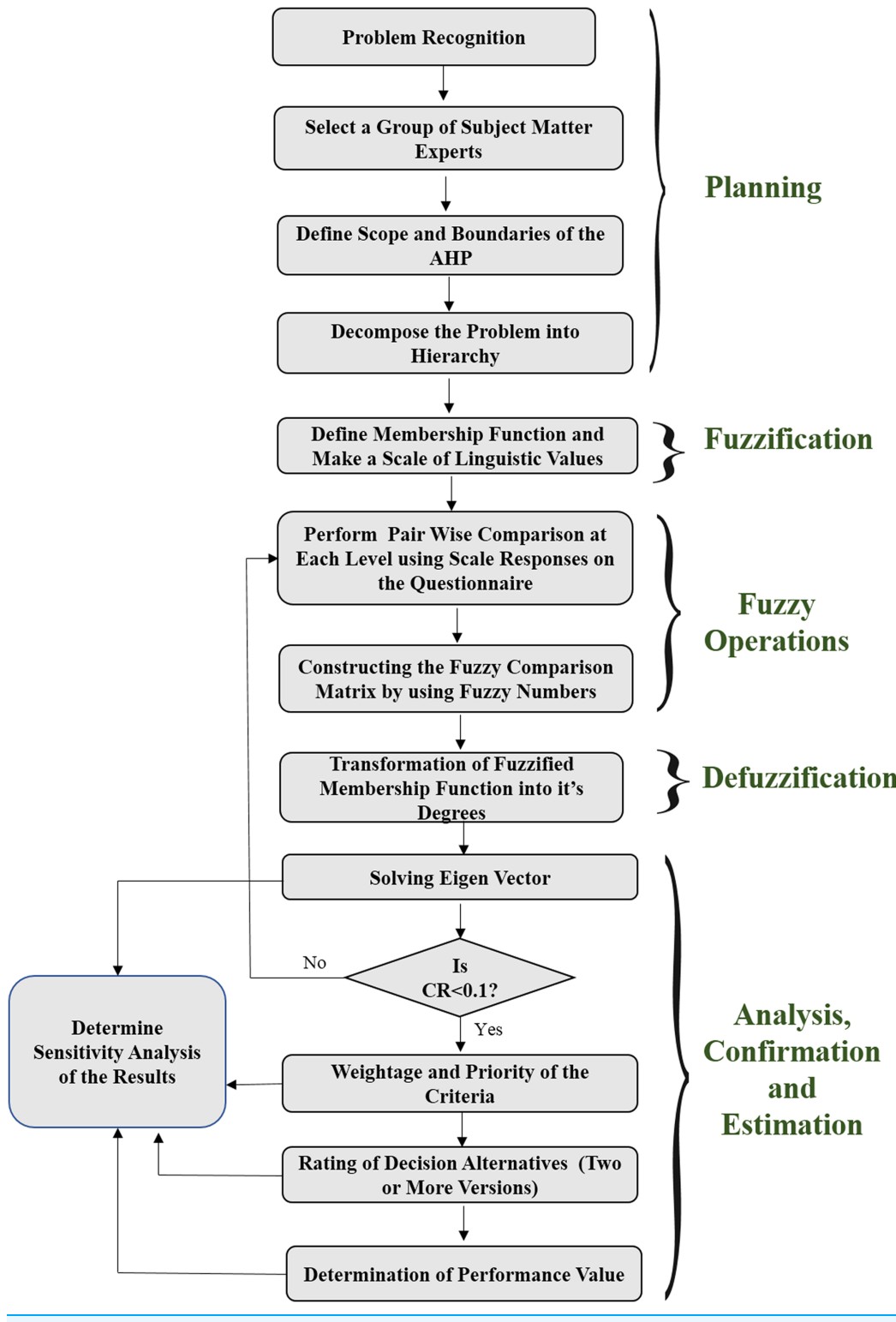

**Figure 1  Flow chart of the implementation through fuzzy AHP method.**

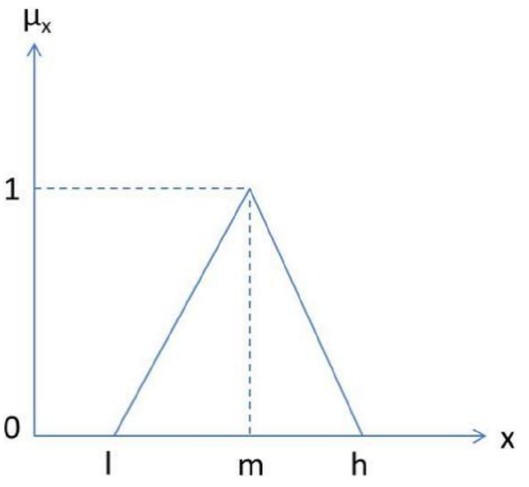

**Figure 2 Triangular fuzzy number.**   

alternatives among the number of options while fuzzy is best in dealing with linguistic variables. That's why Fuzzy AHP has been used in this work for better results.

*Fuzzification phase*

To understand the Fuzzy AHP methodology, researchers have included a short introduction of both methods and hybridization of them. *Saaty (1995)* defines the AHP as a decision method which decomposes a complex multi-criteria decision problem into a hierarchy. The major benefit of AHP is its relative simplicity with which it handles multiple criteria. AHP allows decision makers to mold a complex problem in a hierarchical structure that consists of the goal, aims, sub-objectives, and alternatives. Traditional methods of AHP cannot be used when there is uncertainty in data. To address such uncertainties, the fuzzy set theory was merged into the AHP. *Zadeh (1965)* introduced the fuzzy set theory to deal with the uncertainty due to imprecision and vagueness. A fuzzy set is a class of objects with a graded continuum of membership. Such a set is characterized by a membership function which assigns to each object a membership grade between zero and one. In order to simplify the Fuzzy AHP method for this research from the feasible viewpoints, the Fuzzy AHP based on the fuzzy interval arithmetic with TFNs has been proposed.

In the context of the problem addressed in the present work, Fuzzy AHP has been used for prioritizing security durability attributes. TFN helps the decision maker to make easier decisions (*Chong, Lee & Ling, 2014*). Hence, in this paper TFNs are used as a membership function. Figure 2 depicts a TFN.

In this Fig. 2, $\mu_x$ is denoted as a membership function where $\mu$ denotes membership value of corresponding $x$. The parameters, $l$, $m$, and $h$ denote the smallest possible value, the most promising value, and the largest possible value, respectively, that describes a fuzzy event. Further, a TFN ($\mu_{ij}$) is simply denoted as ($l$, $m$, $h$). The TFN $\mu_{ij}$ is represented in Eq. (1):

$$\mu_{ij} = \left( l_{ij}, \ m_{ij}, \ h_{ij} \right) \tag{1}$$

**Table 1 Corresponding linguistic scale for membership functions.**

| S. no. | Linguistic values | Numeric values | Fuzzified numbers (TFNs) $[a_{ij}]$ | $1/[a_{ij}]$ |
|---|---|---|---|---|
| 1 | Equal important (Eq) | 1 | (1, 1, 1) | (1, 1, 1) |
| 2 | Intermediate value between equal and weakly (E and W) | 2 | (1, 2, 3) | (1/3, 1/2, 1) |
| 3 | Weakly important (WI) | 3 | (2, 3, 4) | (1/4, 1/3, 1/2) |
| 4 | Intermediate value between weakly and essential (W and E) | 4 | (3, 4, 5) | (1/5, 1/4, 1/3) |
| 5 | Essential important (EI) | 5 | (4, 5, 6) | (1/6, 1/5, 1/4) |
| 6 | Intermediate value between essential and very strongly (E and VS) | 6 | (5, 6, 7) | (1/7, 1/6, 1/5) |
| 7 | Very strongly important (VS) | 7 | (6, 7, 8) | (1/8, 1/7, 1/6) |
| 8 | Intermediate value between very strongly and extremely (VS and ES) | 8 | (7, 8, 9) | (1/9, 1/8, 1/7) |
| 9 | Extremely important (ES) | 9 | (7, 9, 9) | (1/9, 1/9, 1/7) |

where $l_{ij} \leq m_{ij} \leq h_{ij}$ and $l_{ij}, m_{ij}, h_{ij} \in \left[\frac{1}{9}, 9\right]$

$$l_{ij} = \min(B_{ijk}),$$

$$m_{ij} = (B_{ij1} \cdot B_{ij2} \ldots \ldots \ldots B_{ijk})1/k \text{ and } h_{ij} = \max(B_{ijk})$$

Where $B_{ijk}$ represents the judgment of experts, $k$ for the importance of two criteria, i.e, $C_i$ and $C_j$. Since each number in the pair-wise comparison matrix represents the subjective opinion of decision makers and is an ambiguous concept, fuzzy numbers work best to consolidate fragmented expert opinions (*Goli, 2013*; *Chong, Lee & Ling, 2014*). *Saaty (1995)* proposed pair-wise comparisons to create the fuzzy judgment matrix, that is, used in the AHP technique and is shown in Eq. (2).

$$
A = [a_{ij}] = \begin{array}{c} \\ C_1 \\ C_2 \\ \vdots \\ C_n \end{array}
\begin{array}{cccc} C_1 & C_2 & \cdots & C_n \end{array}
\left[ \begin{array}{cccc}
1 & a11 & \cdots & a1n \\
1/a21 & 1 & \ldots & a2n \\
\vdots & \vdots & & \\
1/an1 & 1/an2 & \cdots & 1
\end{array} \right]
\tag{2}
$$

Where $i = 1, 2, 3 \ldots \ldots \ldots n$ and $j = 1, 2, 3 \ldots \ldots \ldots \ldots \ldots n$ and $a_{ij} = 1$: when $i = j$; and $a_{ij} = 1/a_{ij}$; when $i \neq j$ where $(a_{ij})$ denotes a TFN for the relative importance of two criteria $C_i$ and $C_j$. Corresponding linguistic scale for membership functions (1–9) is given in Table 1.

Table 1 shows the linguistic values into numeric values and numeric values into TFN values. TFN values may be used for creating the pair-wise comparison matrix of relative criteria, where $a_{ij}$ denotes the relative importance of criteria $i$ comparison with criteria $j$ in the scale. To determine the weights of each set of attributes, this scale is used in the assessment. Further, the decision made by many experts for security durability is summarized as fuzzy pair-wise comparison matrixes. It is also used for characterizing the pair-wise fuzzy judgment matrix which is used in AHP technique. For determining the importance of alternatives, the linguistic rating scale has been shown in Table 2.

**Table 2 Linguistic rating scale.**

| S. no. | Linguistic value | Numeric value of ratings | Fuzzified ratings (TFNs) |
|---|---|---|---|
| 1 | Very low (VL) | 0.1 | (0.0, 0.1, 0.3) |
| 2 | Low (L) | 0.3 | (0.1, 0.3, 0.5) |
| 3 | Medium (M) | 0.5 | (0.3, 0.5, 0.7) |
| 4 | High (H) | 0.7 | (0.5, 0.7, 0.9) |
| 5 | Very high (VH) | 0.9 | (0.7, 0.9, 1.0) |

Table 2 shows the rating scale of 0–1 in scale as 0.1 describes very low, 0.3 describes low (L), and so on. The associated fuzzy values are assigned to every data got from an expert for a particular alternative. The process of assessment starts with collecting data by the different number of experts. Data can be collected in forms of questionnaires, checklist, etc. The data acquired from the decision makers are compared pair-wise to evaluate the relative importance of each criterion, or the degree of preference of one factor to another with respect to each criterion. However, the perception and judgments of human are represented by linguistic and vague for a complex problem (*Saaty, 1995*).

*Fuzzy operations*

After, various linguistic data has been converted into quantitative data into TFN values, to confine the vagueness of the parameters which are related, alternatives such as TFN are used. To aggregate all data into a single form, fuzzy operations are required. If, two TFNs $M_1 = (l_1, m_1, h_1)$ and $M_2 = (l_2, m_2, h_2)$ are given. Then, the rules of operations on them are given below in Eqs. (3)–(5).

$$(l_1, m_1, h_1) + (l_2, m_2, h_2) = (l_1 + l_2, m_1 + m_2, h_1 + h_2) \tag{3}$$

$$(l_1, m_1, h_1) \times (l_2, m_2, h_2) = (l_1 \times l_2, m_1 \times m_2, h_1 \times h_2) \tag{4}$$

$$(l_1, m_1, h_1)^{-1} = \left( \frac{1}{h_1}, \frac{1}{m_1}, \frac{1}{l_1} \right) \tag{5}$$

These fuzzy operations are used in various research areas for decision making in different fields such as decision making, rating, and so on (*Csutora & Buckley, 2001*). Further, it is based on the rationality of uncertainty due to imprecision. A major contribution of fuzzy set theory is its capability of dealing with uncertainty.

*Defuzzification*

After the construction of the comparison matrix, defuzzification is performed to produce a quantifiable value based on the calculated TFN values. The defuzzification method adopted in this work has been derived from *Chong, Lee & Ling (2014)*, *Saaty (1995)*, and *Zadeh (1965)* as formulated in Eqs. (6)–(9) which are commonly referred to as the alpha cut method.

$$\tilde{A} = \left[\tilde{a}_{ij}\right] = \begin{array}{c} \\ C_1 \\ C_2 \\ \vdots \\ C_n \end{array} \begin{array}{cccc} C_1 & C_2 & \cdots & C_n \\ \left[\begin{array}{cccc} 1 & \tilde{a}_{11} & \ldots & \tilde{a}_{1i} \\ 1/\tilde{a}_{21} & 1 & \ldots & \tilde{a}_{2i} \\ \vdots & \vdots & & \vdots \\ 1/\tilde{a}_{j1} & 1/\tilde{a}_{j2} \ldots & & 1 \end{array}\right] \end{array} \tag{6}$$

Matrix $\tilde{A}$ is defined as the defuzzified AHP. Where $[\tilde{a}_{ij}]$ denotes a TFN and shows the relative importance between two criteria $C_i$ and $C_j$. There are different defuzzification methods available in the literature such as centroid, the center of sums, alpha cut, etc. (*Chong, Lee & Ling, 2014*). In this work, researchers used the alpha cut method for defuzzification. Alpha cut enables one to describe a fuzzy set as a composition of crisp sets. Crisp sets simply describe whether an element is either a member of the set or not. To defuzzify fuzzy matrix ($\tilde{A}$) into the crisp matrix ($\rho_{\alpha,\beta}$) is shown in Eqs. (7)–(9) (alpha cut method).

$$\rho_{\alpha,\beta}\left(\tilde{a}_{ij}\right) = \left[\beta \cdot \eta_\alpha\left(l_{ij}\right) + (1 - \beta) \cdot \eta_\alpha\left(h_{ij}\right)\right] \tag{7}$$

where $0 \leq \alpha \leq 1$ and $0 \leq \beta \leq 1$
such that,

$$\eta_\alpha\left(l_{ij}\right) = \left(m_{ij} - l_{ij}\right) \cdot \alpha + l_{ij} \tag{8}$$
$$\eta_\alpha\left(h_{ij}\right) = h_{ij} - \left(h_{ij} - m_{ij}\right) \cdot \alpha \tag{9}$$

In Eqs. (7)–(9), $\eta_\alpha(l_{ij})$ denotes the left-end boundary value of alpha cut for $\tilde{a}_{ij}$ and $\eta_\alpha(l_{ij})$ denotes the right-end boundary value of alpha cut for $\tilde{a}_{ij}$. Further, $\alpha$ and $\beta$ carry the meaning of preferences and risk tolerance of participants. Particularly, $\alpha$ and $\beta$ can be stable or in a fluctuating condition. These two values range between 0 and 1, in such a way that a lesser value indicates greater uncertainty in decision making. Meanwhile, the value of $\alpha$ comes to a stable state when it is increasing particularly. Additionally, $\alpha$ and $\beta$ can be any number between 0 and 1, and analysis is normally set as the following 10 numbers, 0.1, 0.2, up to 0.9 for uncertainty emulation. Since preferences and risk tolerance are not the focus of this contribution, the value of 0.5 for $\alpha$ and $\beta$ is used to represent a balanced value. This indicates that attributes are neither extremely optimistic nor pessimistic about their comparison. Variation due to the value of $\alpha$ and $\beta$ is discussed in the sensitivity analysis section. The single pair-wise comparison matrix is shown in Eq. (10).

$$\rho_{\alpha,\beta}\left(\tilde{A}\right) = \rho_{\alpha,\beta}\left[\tilde{a}_{ij}\right] = \begin{array}{c} \\ C_1 \\ C_2 \\ \vdots \\ C_n \end{array} \begin{array}{cccc} C_1 & C_2 & \cdots & C_n \\ \left[\begin{array}{cccc} 1 & \rho_{\alpha,\beta}(\tilde{a}_{11}) & \ldots & \rho_{\alpha,\beta}(\tilde{a}_{1i}) \\ 1/\rho_{\alpha,\beta}(\tilde{a}_{21}) & 1 & \ldots & \rho_{\alpha,\beta}(\tilde{a}_{2i}) \\ \vdots & \vdots & & \vdots \\ 1/\rho_{\alpha,\beta}(\tilde{a}_{j1}) & 1/\rho_{\alpha,\beta}(\tilde{a}_{j2}) & \ldots & 1 \end{array}\right] \end{array} \tag{10}$$

After defuzzification, to validate the consistency of the matrix, next portion of the section has been discussed.

*Analysis, confirmation, and estimation*

The next step is to determine the eigenvalue and eigenvector of the fuzzy pair-wise comparison matrix. The purpose of calculating the eigenvector is to determine the aggregated weight of particular criteria. Assume that $W$ denotes the eigenvector, $I$ denotesunitary matrix while $\lambda$ denotes the eigenvalue of fuzzy pair-wise comparison matrix $\tilde{A}$ or $[\tilde{a}_{ij}]$.

$$\left[(\rho_{\alpha,\beta} \times \tilde{A}) - \lambda \times I\right]. \quad W = 0 \tag{11}$$

Where $\tilde{A}$ is a fuzzy matrix containing fuzzy numbers of the $\rho_{\alpha,\beta}(\tilde{A})$. Equation (11) is based on the linear transformation of vectors. By applying Eqs. (1)–(11), the weight of particular criteria with respect to all other possible criteria can be acquired. The eigenvectors of associated attributes of security durability were then calculated using Eq. (11) as shown in Eq. (12).

$$\left[\left(\rho_{\alpha,\beta} \times \tilde{A}\right) - \lambda \times I\right]. W = \begin{bmatrix} 1 & \rho\alpha,\beta(\tilde{a}_{11}) & \dots & \rho\alpha,\beta(\tilde{a}_{1i}) \\ 1/\rho\alpha,\beta(\tilde{a}_{21}) & 1 & \dots & \rho\alpha,\beta(\tilde{a}_{2i}) \\ \vdots & \vdots & & \vdots \\ 1/\rho\alpha,\beta(\tilde{a}_{j1}) & 1/\rho\alpha,\beta(\tilde{a}_{j2})\dots & & 1 \end{bmatrix} \tag{12}$$

Multiplying eigenvalue $\lambda$ with unitary matrix $I$ produced an identity matrix that cancels out each other. Thus, the notation $\lambda I$ is discarded in this case. Applying Eqs. (11) and (12) results are shown in Eq. (13).

$$\begin{bmatrix} 1 & \rho_{\alpha,\beta}(\tilde{a}_{11}) & \dots & \rho_{\alpha,\beta}(\tilde{a}_{1i}) \\ 1/\rho_{\alpha,\beta}(\tilde{a}_{21}) & 1 & \dots & \rho_{\alpha,\beta}(\tilde{a}_{2i}) \\ \vdots & \vdots & & \vdots \\ 1/\rho_{\alpha,\beta}(\tilde{a}_{j1}) & 1/\rho_{\alpha,\beta}(\tilde{a}_{j2})\dots & & 1 \end{bmatrix} \times \begin{bmatrix} W1 \\ W2 \\ \vdots \\ W_n \end{bmatrix} = \begin{bmatrix} 0 \\ 0 \\ \vdots \\ 0 \end{bmatrix} \tag{13}$$

The aggregated results in terms of weights are shown in Eq. (13).

In order to control the results of the method, the consistency ratio (CR) for each of the matrixes for the hierarchal structure is calculated with the help of Eq. (14).

$$CR = \frac{CI}{RI} \tag{14}$$

Where consistency index denotes as CI and random index denotes as RI (*Saaty, 1995*). Further, CI is calculated from Eq. (15).

$$CI = \frac{\lambda}{(n-1)} \tag{15}$$

Where $n$ denotes the number of total responses and RI is given by *Saaty (1995)* and given the rank of a matrix as shown in Table 3.

With the help of Eqs. (14) and (15) and Table 3, CR is calculated. If, CR < 0.1, the approximation is accepted and results are evaluated after this with the help of Eq. (13); otherwise, a new comparison matrix is solicited.

| Table 3 Random index. | | | | | | | | | |
|---|---|---|---|---|---|---|---|---|---|
| N | 1 | 2 | 3 | 4 | 5 | 6 | 7 | 8 | 9 |
| Random index (RI) | 0.00 | 0.00 | 0.58 | 0.90 | 1.12 | 1.24 | 1.35 | 1.41 | 1.49 |

After calculating the independent weights, this work evaluates the dependent weights and ranks through the hierarchy and results of the obtainable weights gives some suggestion for developers to improve the security durability life span of software services. To assess the effectiveness of results, this work takes two alternatives (version 1 and version 2). Design of version 1 is original from the organization and design of version 2 is changed according to the priorities. Through the hierarchy, authors estimate the independent and dependent ratings of security durability attributes (for version 1 and version 2, respectively) with the help of Eqs. (1), (3–5), and (7–9). Then, the authors have assessed the security durability of both alternatives. Overall, the security durability is assessed by Eq. (16) (*Chang, Wu & Lin, 2008*).

$$SecurityDurability = R_1 \times W_1 + R_2 \times W_2 + \ldots \ldots R_n \times W_n = \Sigma R_i \times W_i \qquad (16)$$

Where $R$ denotes the rating values, $W$ denotes the weight of associated attribute, and $I$ denotes the number of attributes that affect the security durability. The results clearly underline the impact of the researchers' suggestions and this research work. Further, sensitivity analysis is performed to check the variations on results due to the value of $\alpha$ and $\beta$.

## Security durability assessment

A mechanism for security durability assessment has already been discussed in the previous section. According to the mechanism, firstly, researchers will evaluate the local weights of security durability attributes through Fuzzy AHP technique (fuzzy method) and put the local weights in the hierarchy and will find the most important attributes in the form of ranks and their final weights. After this, the authors will give suggestions/guidelines for the developers to improve the security life span of software services. To evaluate the security durability of software and impact of the suggestions, researchers are taking two versions of BBAU software, that is, version 1 and version 2 where design of version 1 is based on the organizations (called old version) and design of version 2 is modified, according to the given suggestions (called modified version). To assess the best alternative, the ratings of version 1 and version 2 will be evaluated through fuzzy average method (*Kumar et al., 2019*; *Baas & Kwakernaak, 1977*). With the help of weights (also called subjective weights) and ratings (also called objective weights) of the attributes, overall security durability of version 1 and version 2 will be estimated. The step-by-step process of assessment has been shown in the next portion of the section.

### *Evaluating the weights of the attributes*

Through the previous discussion and literature studies, it is found that integrating durability within design may enhance the potential of CIA (*FCW Workshop, 2016*). Hence,

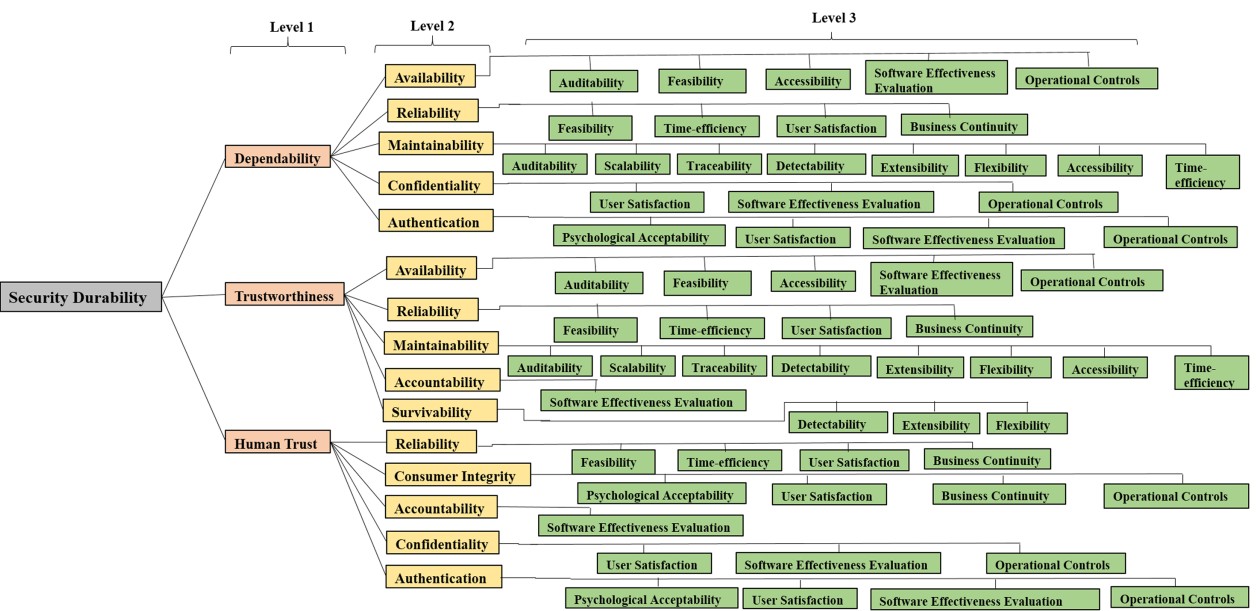

**Figure 3 Hierarchy modeling of security durability attributes.**

firstly establishing a relation between durability and security is important. Security of a software product is durable if it works efficiently for user's satisfaction up to the expected duration. Identification and classification of security durability attributes help to improve security during software development. In order to develop durable as well as secure software, the relationship between security and durability characteristics (at different levels) has been determined in the authors' previous work (*Kumar, Khan & Khan, 2015*). For using the methodology of Fuzzy AHP, these attributes and sub-attributes are converted into a hierarchy that is shown in Fig. 3.

Figure 3 depicts the hierarchical structure of security durability and its attributes which are classified in three levels. At different levels of the hierarchy, the relationship between software quality attributes and software security attributes is shown. Finally, the association of software security attributes with software durability attributes has been shown. An attribute at level 1 affects one or more attributes at the higher level but its effect is not same on them, it may vary. For example, reliability has an impact on dependability, human trust, and trustworthiness as well (*Kumar et al., 2019*), but its impact values are not same in both levels. Further, the hierarchy of attributes helps to differentiate among the impact of the same attribute to the other attribute at the higher level. Among all the attributes, trustworthiness, human trust, and dependability affect the durability directly but many attributes of security affect durability indirectly as well, for example, availability, etc. For the purpose of estimation of security durability, attributes at level 1 are denoted as C1, C2, and C3. Attributes at level 2 are denoted as C11, C12, C13, C14, C15 for C1 and C21, C22 . . . . . . C25 for C2 and C31, C32 . . . . .C35 for C3. Attributes at level 3 are denoted as C111 . . . . . . . . . .C115 for C11 and so on which are shown in Fig. 3.

**Table 4 Aggregated fuzzify pair wise comparison matrix for the first level.**

|  | Dependability (C1) | Trustworthiness (C2) | Human trust (C3) |
|---|---|---|---|
| Dependability (C1) | 1 | 1.3479, 1.8180, 2.3859 | 1.4131, 1.9651, 2.4820 |
| Trustworthiness (C2) | – | 1 | 0.8540, 1.1087, 1.4532 |
| Human trust (C3) | – | – | 1 |

### Construction of pair-wise comparison matrices

Many times, the assessment of different attributes usually fails because of the connection of multiple qualitative criteria. Fuzzy AHP is a suitable evaluation technique capable of handling this kind of problem with uncertain inputs. Fuzzy AHP is capable of handling ambiguous judgmental inputs given by the number of experts and questionnaires collected by judgments of experts. It is also capable of converting qualitative inputs into quantitative results, in form of weight, ranking as well as performance. To evaluate the weights of the security durability attributes, pair-wise comparison matrixes are constructed in the form of questionnaires for each set of attributes and data has been collected by distributing questionnaires to 50 academicians and industry persons of various affiliations. A total of 20 valid replies were used in this research to measure the importance of security durability attributes.

The data collected through expert's opinions have been arranged in the form of decision matrices. Eigenvector method has been used for taking expert's views. Also, repeated data and redundancy has been removed using "data only once" method. Although during calculation, these repetitions have been taken into account as every attribute has a different impact on security durability at different levels of hierarchy. To construct the pair-wise comparison matrices, Table 1 shows a scale in the previous section. This scale is a 9-point scale ranging from 1 to 9, where a greater value represents higher importance. This scale also helped to convert the numerical values into TFN. TFN's can be obtained for computing the fuzzified values of the linguistic terms from the pair-wise judgment matrix. Further, TFN helps the person in making the decision easily. Hence, TFN is used as the membership function in this work.

### Aggregation of pair-wise comparison matrices

With the help of Table 1 and Eqs. (1)–(5) given in the mechanism section, authors converted the numerical values into TFN and aggregated these values. For all sets of attributes of the hierarchy, aggregated pair-wise comparison matrices are shown from Tables 4 to 14.

Table 4 shows the aggregated fuzzify pair-wise comparison matrix of first level attributes including dependability (C1), trustworthiness (C2), and human trust (C3).

Table 5 shows the aggregated fuzzify pair-wise comparison matrix of second level attributes for dependability including availability (C11), reliability (C12), maintainability (C13), confidentiality (C14), and authentication (C15).

Table 6 shows the aggregated fuzzify pair-wise comparison matrix of second level attributes for trustworthiness including availability (C21), reliability (C22), maintainability (C23), accountability (C24), and survivability (C25).

**Table 5 Aggregated fuzzify pair wise comparison matrix for C1 of second level.**

|  | Availability (C11) | Reliability (C12) | Maintainability (C13) | Confidentiality (C14) | Authentication (C15) |
|---|---|---|---|---|---|
| Availability (C11) | 1 | 0.3127, 0.4395, 0.6252 | 0.8733, 0.9012, 0.9465 | 0.2261, 0.2928, 0.4166 | 0.2580, 0.3386, 0.5055 |
| Reliability (C12) | – | 1 | 2.0451, 3.1699, 4.2330 | 0.2665, 0.3657, 0.5911 | 0.6906, 1.0059, 1.5117 |
| Maintainability (C13) | – | – | 1 | 0.3667, 0.5251, 0.9659 | 0.3604, 0.5220, 0.8074 |
| Confidentiality (C14) | – | – | – | 1 | 0.8960, 1.1486, 1.3903 |
| Authentication (C15) | – | – | – | – | 1 |

**Table 6 Aggregated fuzzify pair wise comparison matrix for C2 of second level.**

|  | Availability (C21) | Reliability (C22) | Maintainability (C23) | Accountability (C24) | Survivability (C25) |
|---|---|---|---|---|---|
| Availability (C21) | 1 | 0.5598, 0.8994, 1.3705 | 0.7912, 0.8831, 1.0204 | 0.4956, 0.7029, 0.9330 | 0.4067, 0.5497, 0.7876 |
| Reliability (C22) | – | 1 | 0.8001, 1.2376, 1.7812 | 0.3836, 0.5483, 0.8344 | 0.4876, 0.6710, 0.8900 |
| Maintainability (C23) | – | – | 1 | 0.5966, 0.7093, 0.9095 | 0.2770, 0.3854, 0.6340 |
| Accountability (C24) | – | – | – | 1 | 0.5506, 0.5881, 0.6647 |
| Survivability (C25) | – | – | – | – | 1 |

**Table 7 Aggregated fuzzify pair wise comparison matrix for C3 of second level.**

|  | Reliability (C31) | Consumer integrity (C32) | Accountability (C33) | Confidentiality (C34) | Authentication (C35) |
|---|---|---|---|---|---|
| Reliability (C31) | 1 | 0.9710, 1.2475, 1.6094 | 1.0592, 1.5849, 2.2206 | 0.7733, 1.0118, 1.2881 | 0.7612, 0.9120, 1.0965 |
| Consumer integrity (C32) | – | 1 | 0.6352, 0.9143, 1.3430 | 0.4273, 0.6335, 0.9660 | 0.3476, 0.4900, 0.8734 |
| Accountability (C33) | – | – | 1 | 0.5146, 0.6575, 0.7846 | 0.5213, 0.6597, 0.9191 |
| Confidentiality (C34) | – | – | – | 1 | 0.5562, 0.6448, 0.8122 |
| Authentication (C35) | – | – | – | – | 1 |

**Table 8 Aggregated fuzzify pair wise comparison matrix for C11 of third level.**

|  | Auditability (C111) | Feasibility (C112) | Accessibility (C113) | Software effectiveness evaluation (C114) | Operational controls (C115) |
|---|---|---|---|---|---|
| Auditability (C111) | 1 | 1.8722, 2.5710, 3.2035 | 1.4640, 1.6842, 1.9743 | 1.4461, 2.4385, 3.3865 | 0.4677, 0.5724, 0.7845 |
| Feasibility (C112) | – | 1 | 0.6083, 0.7754, 1.0265 | 0.7708, 0.9504, 1.2361 | 0.1630, 0.1953, 0.2497 |
| Accessibility (C113) | – | – | 1 | 0.7694,10.0502, 1.3553 | 0.2086, 0.2462, 0.3117 |
| Software effectiveness evaluation (C114) | – | – | – | 1 | 0.1956, 0.2283, 0.2903 |
| Operational controls (C115) | – | – | – | – | 1 |

Table 7 shows the aggregated fuzzify pair-wise comparison matrix of second level attributes for human trust including reliability (C31), consumer integrity (C32), accountability (C33), confidentiality (C34), and authentication (C35).

Table 8 shows the aggregated fuzzify pair-wise comparison matrix of third level attributes for availability (related to dependability) including auditability (C111), feasibility (C112), accessibility (C113), software effectiveness evaluation (C114), and operational controls (C115).

**Table 9  Aggregated fuzzify pair wise comparison matrix for the C12 of third level.**

|  | Feasibility (C121) | Time-efficiency (C122) | User satisfaction (C123) | Business continuity (C124) |
|---|---|---|---|---|
| Feasibility (C121) | 1 | 1.7561, 2.3498, 3.0335 | 1.4830, 1.9575, 2.5293 | 1.1284, 1.5543, 1.9884 |
| Time-efficiency (C122) | – | 1 | 0.5695, 0.7860, 1.1555 | 0.5698, 0.7195, 0.9699 |
| User satisfaction (C123) | – | – | 1 | 0.6270, 0.8123, 1.0718 |
| Business continuity (C124) | – | – | – | 1 |

**Table 10  Aggregated fuzzify pair wise comparison matrix for the C13 of third level.**

|  | Auditability (131) | Scalability (132) | Traceability (133) | Detectability (134) | Extensibility (135) | Flexibility (136) | Accessibility (137) | Time-efficiency (138) |
|---|---|---|---|---|---|---|---|---|
| Auditability (131) | 1 | 1.0000, 1.5157, 1.9331 | 0.4896, 0.6372, 1.0000 | 0.4152, 0.5743, 1.0000 | 0.2215, 0.2871, 0.4152 | 0.3146, 0.4610, 0.8705 | 0.6575, 1.1653, 1.6883 | 0.2444, 0.3238, 0.4801 |
| Scalability (132) | – | 1 | 0.5743, 0.6657, 0.8022 | 0.3039, 0.3936, 0.5661 | 0.2679, 0.3521, 0.5176 | 0.1663, 0.1969, 0.2531 | 0.3930, 0.5743, 1.0564 | 0.1692, 0.2076, 0.2759 |
| Traceability (133) | – | – | 1 | 1.0000, 1.3195, 1.5518 | 0.3009, 0.4352, 0.8027 | 0.8027, 0.8705, 1.0000 | 1.2619, 1.8250, 2.4334 | 0.1728, 0.2091, 0.2648 |
| Detectability (134) | – | – | – | 1 | 0.5386, 0.9143, 1.5836 | 0.6083, 1.0592, 1.6829 | 0.7503, 1.3465, 1.9611 | 0.6790, 0.7489, 0.8705 |
| Extensibility (135) | – | – | – | – | 1 | 0.4152, 0.6372, 1.1791 | 0.9465, 1.1095, 1.2457 | 0.2500, 0.3300, 0.5000 |
| Flexibility (136) | – | – | – | – | – | 1 | 1.8881, 2.5508, 3.1697 | 0.8027, 1.0352, 1.3160 |
| Accessibility (137) | – | – | – | – | – | – | 1 | 0.2136, 0.2575, 0.3195 |
| Time-efficiency (138) | – | – | – | – | – | – | – | 1 |

**Table 11  Aggregated fuzzify pair wise comparison matrix for the C14 of third level.**

|  | User satisfaction (C141) | Software effectiveness evaluation (C142) | Operational controls (C143) |
|---|---|---|---|
| User satisfaction (C141) | 1 | 0.6898, 0.8860, 1.1002 | 0.2255, 0.2762, 0.3574 |
| Software effectiveness evaluation (C142) | – | 1 | 0.3051, 0.3892, 0.5609 |
| Operational controls (C143) | – | – | 1 |

Table 9 shows the aggregated fuzzify pair-wise comparison matrix of third level attributes for reliability (related to dependability) including feasibility (C121), time-efficiency (C122), user satisfaction (C123), and business continuity (C124).

Table 10 shows the aggregated fuzzify pair-wise comparison matrix of third level attributes for maintainability (related to dependability) including auditability (C131), scalability (C132), traceability (C133), detectability (C134), extensibility (C135), flexibility (C136), accessibility (C137), and time-efficiency (C138).

Table 11 shows the aggregated fuzzify pair-wise comparison matrix of third level attributes for confidentiality (related to dependability) including user satisfaction (C141), software effectiveness evaluation (C142), and operational controls (C143).

**Table 12 Aggregated fuzzify pair wise comparison matrix for the C15 of third level.**

| | Psychological acceptability C151) | User satisfaction (C152) | Software effectiveness evaluation (C153) | Operational controls (C154) |
|---|---|---|---|---|
| Psychological acceptability (C151) | 1 | 1.0000, 1.3741, 1.7118 | 0.5610, 0.8360, 1.0781 | 0.3040, 0.3766, 0.4723 |
| User satisfaction (C152) | – | 1 | 0.3030, 0.4208, 0.6052 | 0.1916, 0.2303, 0.3001 |
| Software effectiveness evaluation (C153) | – | – | 1 | 0.5138, 0.7959, 1.2032 |
| Operational controls (C154) | – | – | – | 1 |

**Table 13 Aggregated fuzzify pair wise comparison matrix for the C25 of third level.**

| | Detectability (C251) | Extensibility (C252) | Flexibility (C253) |
|---|---|---|---|
| Detectability (C251) | 1 | 0.6950, 0.9502, 1.3457 | 1.1486, 1.4385, 1.6962 |
| Extensibility (C252) | – | 1 | 1.1928, 1.5826, 2.1497 |
| Flexibility (C253) | – | – | 1 |

**Table 14 Aggregated fuzzify pair wise comparison matrix for the C32 of third level.**

| | Psychological acceptability (C321) | User satisfaction (C322) | Business continuity (C323) | Operational controls (C324) |
|---|---|---|---|---|
| Psychological acceptability (C321) | 1 | 1.07810, 1.5990, 2.1130 | 0.8206, 1.1118, 1.6150 | 0.5670, 0.7132, 0.8739 |
| User satisfaction (C322) | – | 1 | 0.3230, 0.4480, 0.6051 | 0.2584, 0.3172, 0.4168 |
| Business continuity (C323) | – | – | 1 | 0.6661, 1.0564, 1.5427 |
| Operational controls (C324) | – | – | – | 1 |

Table 12 shows the aggregated fuzzify pair-wise comparison matrix of third level attributes for authentication (related to dependability) including psychological acceptability (C151), user satisfaction (C152), software effectiveness evaluation (C153), and operational controls (C154).

Due to repeated attributes in the second level, some set of third level attributes are repeated when the set of attributes considered independently. Hence, aggregated fuzzify pair-wise comparison matrixes of third level attributes for C21, C22, and C23 (related to trustworthiness) are same as C11, C12, and C13, respectively. According to hierarchy, accountability (C24) depends only on software effectiveness evaluation (C241) with respect to security durability. So, there is no need of fuzzify pair-wise comparison matrix. Further, aggregated fuzzify pair-wise comparison matrix for the C25 of the third level is shown in Table 13.

Table 13 shows the aggregated fuzzify pair-wise comparison matrix of third level attributes for survivability (related to trustworthiness) including detectability (C251), extensibility (C252), and flexibility (C253).

Table 14 shows the aggregated fuzzify pair-wise comparison matrix of third level attributes for consumer integrity (related to human trust) including psychological acceptability (C321), user satisfaction (C322), business continuity (C323), and operational controls (C324). Again, aggregated fuzzify pair-wise comparison matrixes of third level attributes for C31, C34, and C35 (related to human trust) are same as C12, C14, and C15, respectively.

**Table 15 Local weight of attributes for first level through fuzzy method.**

| | Dependability (C1) | Trustworthiness (C2) | Human trust (C3) | Weights |
|---|---|---|---|---|
| Dependability (C1) | 1 | 1.8425 | 1.9564 | 0.4867 |
| Trustworthiness (C2) | 0.5427 | 1 | 1.1312 | 0.2698 |
| Human trust (C3) | 0.5111 | 0.8840 | 1 | 0.2435 |
| CR = 0.00038 | | | | |

**Table 16 Local weight of attributes for C1 of second level through fuzzy method.**

| | Availability (C11) | Reliability (C12) | Maintainability (C13) | Confidentiality (C14) | Authentication (C15) | Weights |
|---|---|---|---|---|---|---|
| Availability (C11) | 1 | 0.4542 | 0.9056 | 0.3071 | 0.3602 | 0.0946 |
| Reliability (C12) | 2.2017 | 1 | 3.1545 | 0.3973 | 1.0536 | 0.2292 |
| Maintainability (C13) | 1.1042 | 0.31701 | 1 | 0.5957 | 0.5530 | 0.1192 |
| Confidentiality (C14) | 3.2563 | 2.5170 | 1.6787 | 1 | 1.1459 | 0.3233 |
| Authentication (C15) | 2.7762 | 0.9491 | 1.8083 | 0.8727 | 1 | 0.2337 |
| C.R. = 0.0411 | | | | | | |

Further, accountability (C33) depends only on software effectiveness evaluation (C331) with respect to security durability. So, there is no need for fuzzify pair-wise comparison matrix. After the Aggregation of fuzzify pair-wise comparison matrixes, defuzzification process is implemented in the next portion.

### Defuzzification and local weights

Now for getting the linguistic values from the aggregated TFN values, the alpha cut method is used for defuzzification process (*Goli, 2013*). Alpha Cut method is formulated in Eqs. (6)–(9) in the previous section.

All aggregated TFN values that are defuzzified have been shown from the Tables 15 to 25. In this work, α and β are taken equal to 0.5. Where α and β carry the meaning of preferences and risk tolerance of participants. The values of α = 0.5 and β = 0.5 indicated that attributes are neither extremely optimistic nor pessimistic about their comparison. Further, variation in results due to the value of α and β is discussed in the sensitivity analysis section. After defuzzification of pair-wise matrix, CR is calculated with the help of Eqs. (14) and (15) and Table 6 as already discussed in the previous section. To continue the Fuzzy AHP analysis, CR must be acceptable. If CR is less than 0.1, then weights are calculated. Otherwise refined pair-wise matrixes are prepared and the process is repeated again. After verification of the CR value, by applying Eqs. (12) and (13), local weights of security durability attributes are calculated. Tables 15–25 depicts the local weights and CR values for each pair-wise comparison matrix. CR is less than 0.1 for all matrices. This CR value is acceptable to continue Fuzzy AHP analysis.

A local weight shows the level-wise impact of these attributes and is also called independent weight. To evaluate the weights of the security durability attributes throughout the hierarchy, final weights have been calculated in the next portion.

**Table 17 Local weight of attributes for C2 of second level through fuzzy method.**

|  | Availability (C21) | Reliability (C22) | Maintainability (C23) | Accountability (C24) | Survivability (C25) | Weights |
|---|---|---|---|---|---|---|
| Availability (C21) | 1 | 0.9323 | 0.8945 | 0.7086 | 0.5734 | 0.1541 |
| Reliability (C22) | 1.0726 | 1 | 1.2642 | 0.5787 | 0.6647 | 0.1692 |
| Maintainability (C23) | 1.1179 | 0.7910 | 1 | 0.7304 | 0.4205 | 0.1476 |
| Accountability (C24) | 1.4112 | 1.7280 | 1.3691 | 1 | 0.5979 | 0.2214 |
| Survivability (C25) | 1.7440 | 1.5044 | 2.3781 | 1.6725 | 1 | 0.3077 |
| C.R. = 0.0101 | | | | | | |

**Table 18 Local weight of attributes for C3 of second level through fuzzy method.**

|  | Reliability (C31) | Consumer-integrity (C32) | Accountability (C33) | Confidentiality (C34) | Authentication (C35) | Weights |
|---|---|---|---|---|---|---|
| Reliability (C31) | 1 | 1.2689 | 1.6124 | 1.0213 | 0.9204 | 0.2216 |
| Consumer integrity (C32) | 0.7881 | 1 | 1.2693 | 0.6651 | 0.5503 | 0.1596 |
| Accountability (C33) | 0.6202 | 0.7878 | 1 | 0.6536 | 0.6900 | 0.1446 |
| Confidentiality (C34) | 0.9791 | 1.5035 | 1.5300 | 1 | 0.6645 | 0.2115 |
| Authentication (C35) | 1.0865 | 1.8172 | 1.4493 | 1.5049 | 1 | 0.2627 |
| C.R. = 0.0069 | | | | | | |

**Table 19 Local weight of attributes for C11 of third level through fuzzy method.**

|  | Auditability (C111) | Feasibility (C112) | Accessibility (C113) | Software effectiveness evaluation (C114) | Operational controls (C115) | Weights |
|---|---|---|---|---|---|---|
| Auditability (C111) | 1 | 2.5544 | 1.7017 | 2.4274 | 0.5993 | 0.2400 |
| Feasibility (C112) | 0.3915 | 1 | 0.7964 | 0.9769 | 0.2073 | 0.0952 |
| Accessibility (C113) | 0.5876 | 1.2556 | 1 | 1.0563 | 0.2532 | 0.1200 |
| Software effectiveness evaluation (C114) | 0.4120 | 1.0236 | 0.9467 | 1 | 0.2357 | 0.1032 |
| Operational controls (C115) | 1.6686 | 4.8239 | 3.9495 | 4.2427 | 1 | 0.4416 |
| C.R. = 0.0025 | | | | | | |

**Table 20 Local weight of attributes for C12 of third level through fuzzy method.**

|  | Feasibility (C121) | Time-efficiency (C122) | User satisfaction (C123) | Business continuity (C124) | Weights |
|---|---|---|---|---|---|
| Feasibility (C121) | 1 | 2.3723 | 1.9819 | 1.5564 | 0.3905 |
| Time-efficiency (C122) | 0.4215 | 1 | 0.8243 | 0.7447 | 0.1694 |
| User satisfaction (C123) | 0.5046 | 1.2132 | 1 | 0.8309 | 0.2004 |
| Business continuity (C124) | 0.6425 | 1.3428 | 1.2035 | 1 | 0.2397 |
| CR = 0.0006 | | | | | |

### Final weights of each attribute

Final weights are also called dependent weights of security durability throughout the hierarchy. The final weights (dependent weights) of each attribute through hierarchy are shown in Table 26.

**Table 21  Local weight of attributes for C13 of third level through fuzzy method.**

|  | Auditability (131) | Scalability (132) | Traceability (133) | Detectability (134) | Extensibility (135) | Flexibility (136) | Accessibility (137) | Time-efficiency (138) | Weights |
|---|---|---|---|---|---|---|---|---|---|
| Auditability (131) | 1 | 1.4912 | 0.6910 | 0.6410 | 0.3027 | 0.5268 | 1.1691 | 0.3430 | 0.0733 |
| Scalability (132) | 0.6706 | 1 | 0.6770 | 0.4143 | 0.3724 | 0.2033 | 0.6495 | 0.2151 | 0.0497 |
| Traceability (133) | 1.4470 | 1.4771 | 1 | 1.2977 | 0.4935 | 0.8520 | 1.8364 | 0.2140 | 0.1031 |
| Detectability (134) | 1.5600 | 2.4137 | 0.7706 | 1 | 0.9636 | 1.1024 | 1.3511 | 0.7319 | 0.1271 |
| Extensibility (135) | 3.3036 | 2.6853 | 2.0263 | 1.0378 | 1 | 0.7172 | 1.1028 | 0.4350 | 0.1414 |
| Flexibility (136) | 1.8982 | 4.9188 | 1.1737 | 0.9071 | 1.3943 | 1 | 2.3852 | 1.0473 | 0.1729 |
| Accessibility (137) | 0.8554 | 1.5397 | 0.5445 | 0.7401 | 0.90679 | 0.41925 | 1 | 0.2621 | 0.0760 |
| Time-efficiency (138) | 2.9154 | 4.6490 | 4.6729 | 1.36631 | 2.2989 | 0.95484 | 3.8153 | 1 | 0.2565 |

C.R. = 0.0333

**Table 22  Local weight of attributes for C14 of third level through fuzzy method.**

|  | User satisfaction (C141) | Software effectiveness evaluation (C142) | Operational controls (C143) | Weights |
|---|---|---|---|---|
| User satisfaction (C141) | 1 | 0.8905 | 0.2839 | 0.1832 |
| Software effectiveness evaluation (C142) | 1.1230 | 1 | 0.4111 | 0.2239 |
| Operational controls (C143) | 3.5224 | 2.4325 | 1 | 0.5929 |

C.R. = 0.0062

**Table 23  Local weight of attributes for C15 of third level through fuzzy method.**

|  | Psychological acceptability (C151) | User satisfaction (C152) | Software effectiveness evaluation (C153) | Operational controls (C154) | Weights |
|---|---|---|---|---|---|
| Psychological acceptability (C151) | 1 | 1.3651 | 0.8278 | 0.3824 | 0.1811 |
| User satisfaction (C152) | 0.7325 | 1 | 0.4375 | 0.2381 | 0.1167 |
| Software effectiveness evaluation (C153) | 1.2080 | 2.2857 | 1 | 0.8272 | 0.2757 |
| Operational controls (C154) | 2.6151 | 4.1999 | 1.2089 | 1 | 0.4265 |

C.R. = 0.0151

**Table 24  Local weight of attributes for C25 of third level through fuzzy method.**

|  | Detectability (C251) | Extensibility (C252) | Flexibility (C253) | Weights |
|---|---|---|---|---|
| Detectability (C251) | 1 | 0.9853 | 1.3578 | 0.3611 |
| Extensibility (C252) | 1.0149 | 1 | 1.6269 | 0.3873 |
| Flexibility (C253) | 0.7365 | 0.6147 | 1 | 0.2516 |

C.R. = 0.0026

The hierarchical structure related to security durability attributes is helpful in building the effective security design of software. The decomposition of security durability attributes has been considered in three levels viz., level 1, level 2, and level 3. Based on the results, the rank of each attribute is obtained at level 1, 2, and 3.

**Table 25 Local weight of attributes for C32 of third level through fuzzy method.**

|  | Psychological acceptability (C321) | User satisfaction (C322) | Business continuity (C323) | Operational controls (C324) | Weights |
|---|---|---|---|---|---|
| Psychological acceptability (C321) | 1 | 1.5973 | 1.1648 | 0.7168 | 0.2543 |
| User satisfaction (C322) | 0.6261 | 1 | 0.4561 | 0.3274 | 0.1302 |
| Business continuity (C323) | 0.8585 |  | 1 | 1.0804 | 0.2829 |
| Operational controls (C324) | 1.3951 | 3.0544 | 0.9256 | 1 | 0.3326 |

C.R. = 0.0187

**Table 26 The final weights of each criterion through hierarchy.**

| The first level | The weight of the first level | The second level | The local weight of the second level | The final weight of the second level | The third level | The local weight of the third level | The (global) final weight of the third level |
|---|---|---|---|---|---|---|---|
| C1 | 0.4867 | C11 | 0.0946 | 0.046 | C111 | 0.2400 | 0.011 |
|  |  |  |  |  | C112 | 0.0952 | 0.004 |
|  |  |  |  |  | C113 | 0.1200 | 0.006 |
|  |  |  |  |  | C114 | 0.1032 | 0.005 |
|  |  |  |  |  | C115 | 0.4416 | 0.020 |
|  |  | C12 | 0.2292 | 0.112 | C121 | 0.3905 | 0.044 |
|  |  |  |  |  | C122 | 0.1694 | 0.019 |
|  |  |  |  |  | C123 | 0.2004 | 0.022 |
|  |  |  |  |  | C124 | 0.2397 | 0.027 |
|  |  | C13 | 0.1192 | 0.058 | C131 | 0.0733 | 0.004 |
|  |  |  |  |  | C132 | 0.0497 | 0.003 |
|  |  |  |  |  | C133 | 0.1031 | 0.006 |
|  |  |  |  |  | C134 | 0.1271 | 0.007 |
|  |  |  |  |  | C135 | 0.1414 | 0.008 |
|  |  |  |  |  | C136 | 0.1729 | 0.010 |
|  |  |  |  |  | C137 | 0.0760 | 0.004 |
|  |  |  |  |  | C138 | 0.2565 | 0.015 |
|  |  | C14 | 0.3233 | 0.157 | C141 | 0.1832 | 0.029 |
|  |  |  |  |  | C142 | 0.2239 | 0.035 |
|  |  |  |  |  | C143 | 0.5929 | 0.093 |
|  |  | C15 | 0.2337 | 0.114 | C151 | 0.1811 | 0.021 |
|  |  |  |  |  | C152 | 0.1167 | 0.013 |
|  |  |  |  |  | C153 | 0.2757 | 0.031 |
|  |  |  |  |  | C154 | 0.4265 | 0.049 |
| C2 | 0.2698 | C21 | 0.1541 | 0.042 | C211 | 0.2400 | 0.010 |
|  |  |  |  |  | C212 | 0.0952 | 0.004 |
|  |  |  |  |  | C213 | 0.1200 | 0.005 |
|  |  |  |  |  | C214 | 0.1032 | 0.004 |
|  |  |  |  |  | C215 | 0.4416 | 0.018 |

(Continued)

| The first level | The weight of the first level | The second level | The local weight of the second level | The final weight of the second level | The third level | The local weight of the third level | The (global) final weight of the third level |
|---|---|---|---|---|---|---|---|
| | | C22 | 0.1692 | 0.046 | C221 | 0.3905 | 0.018 |
| | | | | | C222 | 0.1694 | 0.008 |
| | | | | | C223 | 0.2004 | 0.009 |
| | | | | | C224 | 0.2397 | 0.011 |
| | | C23 | 0.1476 | 0.040 | C231 | 0.0733 | 0.003 |
| | | | | | C232 | 0.0497 | 0.002 |
| | | | | | C233 | 0.1031 | 0.004 |
| | | | | | C234 | 0.1271 | 0.005 |
| | | | | | C235 | 0.1414 | 0.006 |
| | | | | | C236 | 0.1729 | 0.007 |
| | | | | | C237 | 0.0760 | 0.003 |
| | | | | | C238 | 0.2565 | 0.010 |
| | | C24 | 0.2214 | 0.060 | C241 | – | 0.060 |
| | | C25 | 0.3077 | 0.083 | C251 | 0.3611 | 0.030 |
| | | | | | C252 | 0.3873 | 0.032 |
| | | | | | C253 | 0.2516 | 0.021 |
| C3 | 0.2435 | C31 | 0.2216 | 0.054 | C311 | 0.3905 | 0.021 |
| | | | | | C312 | 0.1694 | 0.009 |
| | | | | | C313 | 0.2004 | 0.011 |
| | | | | | C314 | 0.2397 | 0.013 |
| | | C32 | 0.1596 | 0.039 | C321 | 0.2543 | 0.010 |
| | | | | | C322 | 0.1302 | 0.005 |
| | | | | | C323 | 0.2829 | 0.011 |
| | | | | | C324 | 0.3326 | 0.013 |
| | | C33 | 0.1446 | 0.035 | C331 | – | 0.035 |
| | | C34 | 0.2115 | 0.052 | C341 | 0.1832 | 0.009 |
| | | | | | C342 | 0.2239 | 0.012 |
| | | | | | C343 | 0.5929 | 0.031 |
| | | C35 | 0.2627 | 0.064 | C351 | 0.1811 | 0.012 |
| | | | | | C352 | 0.1167 | 0.007 |
| | | | | | C353 | 0.2757 | 0.018 |
| | | | | | C354 | 0.4265 | 0.027 |

On the basis of final weights, evaluation of the ranks of each attribute for improving security durability/security life span of software is illustrated. The required security durability attributes are extracted from Fig. 3 and Table 26 shows the importance of each attribute throughout the hierarchy in the form of priorities. Repeated attributes of level 2 and level 3 are removed and Figs. 4 and 5 show the final priorities of security durability attributes at level 2 and level 3.

| Second Level Characteristics | The final weight of the second level | Final Ranks of the Second Level |
|---|---|---|
| Availability | 0.046 | 10 |
| Reliability | 0.112 | 3 |
| Maintainability | 0.058 | 7 |
| Confidentiality | 0.157 | 1 |
| Authentication | 0.114 | 2 |
| ~~Availability~~ | ~~0.042~~ | ~~12~~ |
| ~~Reliability~~ | ~~0.046~~ | ~~11~~ |
| ~~Maintainability~~ | ~~0.040~~ | ~~13~~ |
| Accountability | 0.060 | 6 |
| Survivability | 0.083 | 4 |
| ~~Reliability~~ | ~~0.054~~ | ~~8~~ |
| Consumer Integrity | 0.039 | 14 |
| ~~Accountability~~ | ~~0.035~~ | ~~15~~ |
| ~~Confidentiality~~ | ~~0.052~~ | ~~9~~ |
| ~~Authentication~~ | ~~0.064~~ | ~~5~~ |

**Set of Attributes without Repetition** →

| Priority | Characteristics of Level 2 |
|---|---|
| 1 | Confidentiality |
| 2 | Authentication |
| 3 | Reliability |
| 4 | Survivability |
| 5 | Accountability |
| 6 | Maintainability |
| 7 | Availability |
| 8 | Consumer Integrity |

**Figure 4 Second level attributes without repetition.**

Figures 4 and 5 show the final priorities of security durability attributes at level 2 and level 3 after removing the repeated attributes. These priorities will help toward creating the development suggestions/guidelines.

### Procedure for improving security durability of software

The purpose of this research work is to enhance the security durability of software based on the suggestions and guidelines proposed by the authors. The suggestions or guidelines inferred from the assessment will surely help the developers to improve the security durability of software during its development. To produce any guidelines for developers related to design, it is important to consider properties of the design.

Object-oriented design properties are measured using its corresponding security metrics (*Goli, 2013*). Further, object-oriented security metrics are useless if they are not mapped to security durability parameters. There are numerous security metric suites available to predict security of the software namely vulnerable association of an object-oriented design (*Chowdhury & Zulkernine, 2010*), security requirements statistics (*Abbadi, 2011*), number of design stage security errors (*Siddiqui, 2017*), critical class coupling (CCC) (*Yadav, Sunil & Uttpal, 2014*), critical class extensibility (CCE) (*Mohammed & Taha, 2016*), critical super class propagation (CSP) (*Chowdhury & Zulkernine, 2010*), classified method inheritance (CMI) (*Alshammari, 2011*), classified attributes inheritance (CAI) (*Abbadi, 2011*), critical design propagation (CDP) (*Yadav, Sunil & Uttpal, 2014*), classified instance data accessibility (CIDA) (*Mohammed & Taha, 2016*), classified methods weight (CMW), and many more (*Alshammari, 2011*). The names specified above are security metrics for the design phase. These metrics are specifically

| Third Level Characteristics | The Final Weight of the Third Level | Final Ranks of the Third Level |
|---|---|---|
| Auditability | 0.011 | 29 |
| Feasibility | 0.004 | 52 |
| Accessibility | 0.006 | 45 |
| Software Effective Evaluation | 0.005 | 48 |
| Operational Controls | 0.020 | 18 |
| Feasibility | 0.044 | 4 |
| Time-Efficiency | 0.019 | 19 |
| User Satisfaction | 0.022 | 14 |
| Business Continuity | 0.027 | 12 |
| Auditability | 0.004 | 53 |
| Scalability | 0.003 | 58 |
| Traceability | 0.006 | 46 |
| Detectability | 0.007 | 42 |
| Extensibility | 0.008 | 40 |
| Flexibility | 0.010 | 33 |
| Accessibility | 0.004 | 54 |
| Time-Efficiency | 0.015 | 23 |
| User Satisfaction | 0.029 | 11 |
| Software Effective Evaluation | 0.035 | 5 |
| Operational Controls | 0.093 | 1 |
| Psychological Acceptability | 0.021 | 15 |
| User Satisfaction | 0.013 | 24 |
| Software Effective Evaluation | 0.031 | 8 |
| Operational Controls | 0.049 | 3 |
| Auditability | 0.010 | 34 |
| Feasibility | 0.004 | 55 |
| Accessibility | 0.005 | 49 |
| Software Effective Evaluation | 0.004 | 56 |
| Operational Controls | 0.018 | 20 |
| Feasibility | 0.018 | 21 |
| Time-Efficiency | 0.008 | 41 |
| User Satisfaction | 0.009 | 37 |
| Business Continuity | 0.011 | 30 |
| Auditability | 0.003 | 59 |
| Scalability | 0.002 | 61 |
| Traceability | 0.004 | 57 |
| Detectability | 0.005 | 50 |
| Extensibility | 0.006 | 47 |
| Flexibility | 0.007 | 43 |
| Accessibility | 0.003 | 60 |
| Time-Efficiency | 0.010 | 35 |
| Software Effective Evaluation | 0.060 | 2 |
| Detectability | 0.030 | 10 |
| Extensibility | 0.032 | 7 |
| Flexibility | 0.021 | 16 |
| Feasibility | 0.021 | 17 |
| Time-Efficiency | 0.009 | 38 |
| User Satisfaction | 0.011 | 31 |
| Business Continuity | 0.013 | 25 |
| Psychological Acceptability | 0.010 | 36 |
| User Satisfaction | 0.005 | 51 |
| Business Continuity | 0.011 | 32 |
| Operational Controls | 0.013 | 26 |
| Software Effective Evaluation | 0.035 | 6 |
| User Satisfaction | 0.009 | 39 |
| Software Effective Evaluation | 0.012 | 27 |
| Operational Controls | 0.031 | 9 |
| Psychological Acceptability | 0.012 | 28 |
| User Satisfaction | 0.007 | 44 |
| Software Effective Evaluation | 0.018 | 22 |
| Operational Controls | 0.027 | 13 |

**Set of Attributes without Repetition** →

| Priority | Characteristics of Level 3 |
|---|---|
| 1 | Operational Controls |
| 2 | Software Effectiveness Evaluation |
| 3 | Feasibility |
| 4 | User Satisfaction |
| 5 | Time-efficiency |
| 6 | Auditability |
| 7 | Psychological Acceptability |
| 8 | Business Continuity |
| 9 | Accessibility |
| 10 | Extensibility |
| 11 | Flexibility |
| 12 | Detectability |
| 13 | Scalability |
| 14 | Traceability |

**Figure 5 Third level attributes without repetition.**

used for measuring the impact of the properties. For example, to measure the coupling of classes, CCC is used by most of the practitioners (*Alshammari, 2011*).

Most of the design properties have positive impact on security attributes including service-oriented design and object-oriented design, etc. (*Siddiqui, 2017*). On the other hand, each design strategy has its own positive and negative impacts on security services of software. In this work, researchers suggest only eight security metrics to developers that

may be helpful for measuring and achieving the priorities of third level factors including CCC, CCE, CSP, CMI, CAI, CDP, CIDA, and CMW.

Through the impact of third level priorities, second level, first level, and overall security durability are measured and achieved. Security durability attributes (third level) affect many design attributes and impact of these attributes may be helpful for assessment through suggested security metrics as:

- Auditability affects design properties such as reusability (*Kumar et al., 2019*), discoverability (*Baas & Kwakernaak, 1977*), design by contract (*Chowdhury & Zulkernine, 2010*), and design size (*Baas & Kwakernaak, 1977*). With the help of CMI and CAI metrics, affected design properties of auditability may be measured and improved (*Abbadi, 2011*). Further, CMI measures the ratio between a number of classified methods and the total number of classified methods and CAI measures the ratio between numbers of classified attributes and the total number of classified attributes.

- Scalability affects design properties such as coupling (*Kumar et al., 2019*) and reusability (*Baas & Kwakernaak, 1977*). With the help of CCC and CMI metrics, affected design properties of scalability may be measured and improved (*Mohammed & Taha, 2016*). Further, CCC helps to measure the ratio between the numbers of all classes linked with classified attributes.

- Feasibility affects design properties such as reusability (*Alshammari, 2011*) and discoverability (*Mohammed & Taha, 2016*). With the help of CAI and CMI metrics, affected design properties of feasibility may be measured and improved.

- Traceability affects design properties such as coupling (*Baas & Kwakernaak, 1977*), abstraction (*Alshammari, 2011*), and discoverability. With the help of CCC and CSP metrics, affected design properties of traceability may be measured and improved (*Alshammari, 2011*). Further, CSP helps to measure the ratio between the numbers of critical super classes and a total number of critical classes in an inheritance hierarchy; and also helps to implement the abstraction.

- Detectability affects design properties such as autonomy (*Chowdhury & Zulkernine, 2010*), discoverability (*Abbadi, 2011*), and cohesion. With the help of CCE metric, affected design properties of detectability may be measured and improved (*Abbadi, 2011*). Further, CCE helps to measure the ratio between numbers of non-finalized classes in design with the critical classes in that design.

- Accessibility affects design properties such as complexity (*Siddiqui, 2017*) and design size (*Yadav, Sunil & Uttpal, 2014*). With the help of CDP and CIDA metrics, affected design properties of accessibility may be measured and improved (*Mohammed & Taha, 2016*). Further, CDP measures the ratio between the number of critical classes and a total number of classes in design and measures the impact of the size of a certain design on security. CIDA is helpful to measure the ratio between the number of classified instance public attributes and a total number of classified attributes in a class (*Hoehl, 2013*). It also measures the impact of the size of a certain design on security.

- Time-efficiency affects design properties such as design size (*Chowdhury & Zulkernine, 2010*) and reusability (*Abbadi, 2011*). With the help of CMI and CAI metrics, affected design properties of time-efficiency may be measured and improved.

- Extensibility affects design properties such as complexity (*Baas & Kwakernaak, 1977*) and reusability (*Yadav, Sunil & Uttpal, 2014*). With the help of CMI and CAI metrics, affected design properties of extensibility may be measured and improved.

- Psychological acceptability affects design properties such as abstraction (*Mohammed & Taha, 2016*), design by contract (*Sommardahl & Durable Software, 2013*) and cohesion (*Baas & Kwakernaak, 1977*). With the help of CSP metric, affected design properties of psychological acceptability may be measured and improved.

- User satisfaction affects design properties such as abstraction (*Alshammari, 2011*) and autonomy (*Mohammed & Taha, 2016*). With the help of CSP and CCE metrics, affected design properties of user satisfaction may be measured and improved.

- Software effectiveness evaluation affects design properties such as abstraction (*Baas & Kwakernaak, 1977*) and coupling (*Abbadi, 2011*). With the help of CCE, CMI, CAI, and CSP metrics, affected design properties of software effectiveness evaluation may be measured and improved.

- Business continuity affects design properties such as coupling and cohesion (*Chowdhury & Zulkernine, 2010*). With the help of CCC and CMW metrics, affected design properties of business continuity may be measured and improved.

- Flexibility affects design properties such as coupling (*Siddiqui, 2017*) and statelessness (*Yadav, Sunil & Uttpal, 2014*). With the help of CMW, CDP, and CCC metrics affected design properties of flexibility may be measured and improved (*Mohammed & Taha, 2016*). Further, CMW helps to measure the ratio between the numbers of classified methods and a total number of methods in a given class. CDP measures the ratio between the number of critical classes and a total number of classes, and also helps to measure the impact of the size of a certain design on security.

- Also, operational controls affect design properties such as coupling (*Kumar et al., 2019*) and statelessness (*Chowdhury & Zulkernine, 2010*). With the help of CMW, CDP, and CCC metrics affected design properties of operational controls may be measured and improved.

Through the measurement of third level attributes, the impact of second level attributes of security durability may be measured. Further, to measure and improve the impact of second level attributes, the following are the referrals:

- Confidentiality is affected by third level attributes including user satisfaction, software effective evaluation, and operational controls. With the help of the metrics of design properties for these attributes, the impact of confidentiality may be measured and improved.

- Authentication is affected by third level attributes including psychological acceptability, user satisfaction, software effectiveness evaluation, and operational controls. With the

help of the metrics of design properties for these attributes, the impact of authentication may be measured and improved.

- Reliability is affected by third level attributes including feasibility, time-efficiency, user satisfaction, and business continuity. With the help of the metrics of design properties for these attributes, the impact of reliability may be measured and improved.
- Survivability is affected by third level attributes including detectability, extensibility, and flexibility. With the help of the metrics of design properties for these attributes, the impact of survivability may be measured and improved.

Through the measurement of second level attributes, the impact of first level attributes of security durability may be measured. Further, to measure and improve the impact of first level attributes, the following are the referrals:

- Dependability is affected by second level attributes including availability, reliability, maintainability, confidentiality, and authentication. With the help of the impact of these second level attributes, the impact of dependability may be measured and improved.
- Trustworthiness is affected by second level attributes including availability, reliability, maintainability, accountability, and survivability. With the help of the impact of these second level attributes, the impact of trustworthiness may be measured and improved.
- Human trust is affected by second level attributes including reliability, consumer integrity, accountability, confidentiality, and authentication. With the help of the impact of these second level attributes, the impact of human trust may be measured and improved.

With the help of given final priorities of level 1, 2, and 3 and above discussion, developers should focus on enhancing the high prioritized attributes. Measurement through the metrics is necessary for enhancing the impact of these attributes on overall security durability of software services. Further, recommendations for better implementation and improvement are descriptively given below:

- Improve security durability awareness among developers by adequate education and training to achieve sound security durability culture in the organizational environment during the use of software services.
- The economic aspect of security life span should be clearly understood and addressed as one of the important factors for the organization in the recent information era.
- Periodically review the performance of security durability policy implementations using the MCDM techniques because these techniques hail from academia as well as the software industry so as to realize the real-world practices.
- The development guidelines that have a positive effect on the highest priority security durability attribute, which in this case, dependability, must be gathered.
- On the basis of assessment, security metric for dependability should be prepared and calculated.

- Focus at dependability, human trust, and trustworthiness which are important factors for the security durability of software services.
- Importance of level 1, level 2, and level 3 attributes must be followed by developers.
- In level 2, confidentiality, authentication, and reliability are more desirable attributes and necessary attributes amongst all the other attributes of security durability.
- In level 3, operational controls, software effectiveness evaluation, and feasibility are more essential and required attribute amongst all the other attributes of security durability.

To analyze the impact of given priorities, suggestions and recommendations, researchers evaluated the performance of security durability in both subjective and objective perspectives. Further, subjective assessment has been done in the previous portion of this section. To evaluate the objective assessment, this work is taking two alternatives of BBAU software, i.e., version 1 and version 2. The process is discussed in the next portion.

### Ratings of attributes

A rating is the evaluation of something, in terms of quality, quantity, or some combination of both. According to *Oxford dictionary* "Rating is a classification of something based on a comparative assessment of their quality, standard, or performance" (*Lexico, 2018*).

To evaluate the objective weights, researchers have taken the ratings of security durability attributes from the development team for BBAU software including version 1 and version 2. Old design of the software is called version 1 and modified design of the software is called version 2. According to the given priorities and recommendations, the suggested metrics will be helpful in modifying the design.

The suggested metrics may be helpful in achieving the priorities attained and reform the security design of software. To measure the impact of security durability attributes for version 1 and version 2, authors converted the linguistic values into numerical values with the help of rating scale Table 2 and fuzzy aggregation method was used to evaluate the ratings (also called objective weights). Further, the fuzzy aggregation method was enlisted in various research areas for decision making, rating, and so on (*Kumar et al., 2019*; *Baas & Kwakernaak, 1977*). The next portion discusses fuzzifying and aggregate of the ratings.

### Fuzzified average ratings

Ratings of security durability attributes are collected at level 1, level 2, and level 3. With the help of rating scale Table 2, linguistic values were converted into numerical values and numerical values into TFN. To confine the vagueness of the parameters, which are related to alternatives, TFN is used (*Chong, Lee & Ling, 2014*). With the help of Eqs. (1) and (3–5), fuzzified average ratings are evaluated. Table 27 shows the fuzzified average ratings of security durability attributes for version 1 and version 2.

Table 27 shows the fuzzified average ratings of security durability attributes (attributes of level 1, level 2, and level 3) for version 1 and version 2. Local ratings of security durability attribute for version 1 and version 2 has been evaluated in the next portion.

| Table 27 | Fuzzified average ratings. | | |
|---|---|---|---|
| S. no. | Characteristics of level 1 | Old version (version 1) | Modified version (version 2) |
| 1 | Dependability | 0.445, 0.615, 0.755 | 0.59, 0.79, 0.95 |
| 2 | Trustworthiness | 0.455, 0.64, 0.74 | 0.64, 0.84, 0.97 |
| 3 | Human trust | 0.44, 0.60, 0.74 | 0.62, 0.82, 0.96 |
| S. no. | Characteristics of level 2 | | |
| 1 | Reliability | 0.53, 0.72, 0.865 | 0.62, 0.81, 0.94 |
| 2 | Availability | 0.46, 0.63, 0.775 | 0.63, 0.82, 0.94 |
| 3 | Authentication | 0.38, 0.55, 0.71 | 0.67, 0.85, 0.95 |
| 4 | Maintainability | 0.445, 0.635, 0.79 | 0.65, 0.84, 0.95 |
| 5 | Confidentiality | 0.56, 0.72, 0.835 | 0.51, 0.70, 0.86 |
| 6 | Accountability | 0.445, 0.615, 0.765 | 0.64, 0.83, 0.95 |
| 7 | Consumer integrity | 0.46, 0.635, 0.78 | 0.73, 0.90, 0.99 |
| 8 | Survivability | 0.495, 0.68, 0.83 | 0.69, 0.87, 0.98 |
| S. no. | Characteristics of level 3 | | |
| 1 | Software effectiveness evaluation | 0.66, 0.60, 0.875 | 0.61, 0.75, 0.93 |
| 2 | User satisfaction | 0.64, 0.81, 0.935 | 0.52, 0.64, 0.84 |
| 3 | Feasibility | 0.49, 0.57, 0.835 | 0.53, 0.65, 0.89 |
| 4 | Operational controls | 0.75, 0.67, 0.985 | 0.66, 0.78, 0.97 |
| 5 | Time-efficiency | 0.35, 0.52, 0.77 | 0.69, 0.85, 0.99 |
| 6 | Auditability | 0.56, 0.6, 0.875 | 0.47, 0.58, 0.83 |
| 7 | Psychological acceptability | 0.43, 0.58, 0.90 | 0.61, 0.72, 0.96 |
| 8 | Business continuity | 0.42, 0.57, 0.905 | 0.52, 0.57, 0.90 |
| 9 | Accessibility | 0.49, 0.61, 0.795 | 0.50, 0.61, 0.84 |
| 10 | Extensibility | 0.44, 0.60, 0.89 | 0.46, 0.56, 0.82 |
| 11 | Flexibility | 0.50, 0.66, 0.84 | 0.43, 0.54, 0.79 |
| 12 | Detectability | 0.51, 0.56, 0.83 | 0.49, 0.59, 0.85 |
| 13 | Scalability | 0.46, 0.62, 0.895 | 0.51, 0.66, 0.85 |
| 14 | Traceability | 0.40, 0.57, 0.845 | 0.49, 0.57, 0.87 |

### Defuzzification and local ratings

With the help of Eqs. (7)–(9), local ratings of security durability attributes are estimated. These local ratings are also called independent ratings. Further, Table 28 maps the local ratings for version 1 and version 2.

Table 28 shows the local ratings of security durability attributes for level 1, level 2, and level 3, respectively. Further, local ratings profile the level-wise impact of these attributes for version 1 and version 2 and are also called independent ratings. To evaluate the impact of the security durability attributes throughout the hierarchy, final ratings are calculated in next portion.

### Final rating of each attribute

Table 28 above shows the independent ratings of every attribute at level 1, 2, and 3. Next step in this row is to calculate the final ratings of attributes according to their place in the

**Table 28 Local rating of the attributes for level 1, 2 and 3.**

| S. no. | Characteristics of level 1 | Old version (version 1) | Modified version (version 2) |
|---|---|---|---|
| 1 | Dependability | 0.608 | 0.78 |
| 2 | Trustworthiness | 0.619 | 0.82 |
| 3 | Human Trust | 0.595 | 0.81 |
| **S. no.** | **Characteristics of level 2** | | |
| 1 | Reliability | 0.709 | 0.79 |
| 2 | Availability | 0.624 | 0.80 |
| 3 | Authentication | 0.548 | 0.83 |
| 4 | Maintainability | 0.626 | 0.82 |
| 5 | Confidentiality | 0.709 | 0.69 |
| 6 | Accountability | 0.610 | 0.81 |
| 7 | Consumer integrity | 0.628 | 0.88 |
| 8 | Survivability | 0.671 | 0.85 |
| **S. no.** | **Characteristics of level 3** | | |
| 1 | Software effectiveness evaluation | 0.626 | 0.76 |
| 2 | User satisfaction | 0.799 | 0.66 |
| 3 | Feasibility | 0.616 | 0.68 |
| 4 | Operational controls | 0.769 | 0.79 |
| 5 | Time-efficiency | 0.540 | 0.84 |
| 6 | Auditability | 0.659 | 0.61 |
| 7 | Psychological acceptability | 0.623 | 0.75 |
| 8 | Business continuity | 0.616 | 0.64 |
| 9 | Accessibility | 0.626 | 0.64 |
| 10 | Extensibility | 0.633 | 0.60 |
| 11 | Flexibility | 0.665 | 0.58 |
| 12 | Detectability | 0.615 | 0.63 |
| 13 | Scalability | 0.649 | 0.67 |
| 14 | Traceability | 0.596 | 0.62 |

hierarchy. For calculating the final ratings, the lower level ratings are multiplied to the higher level ratings. Table 29 shows the final ratings of each attribute through the fuzzy method.

Many attributes at level 2 and level 3 are repeated but their impact on its higher level attributes is different. With the help of hierarchy, dependent ratings are evaluated but there are different impacts of the same attribute. With the help of final ratings and weights, security durability of software is estimated for version 1 and version 2 in the next portion.

## RESULTS

### Assessment of security durability

From Eq. (16), security durability is assessed for two alternatives, i.e., version 1 and version 2 with the help of final ratings ($R_i$) and weights ($W_i$) of attributes. Overall security durability is shown in Table 30.

**Table 29 Final ratings of each attribute.**

| The first level | The ratings of durability factors of the first level | | The second level | Local ratings of the second level | | The final ratings of the second level | | The level of the third level | The local ratings of the third level | | The final ratings of the third level | |
|---|---|---|---|---|---|---|---|---|---|---|---|---|
| | Version 1 | Version 2 | | Version 1 | Version 2 | Version 1 | Version 2 | | Version 1 | Version 2 | Version 1 | Version 2 |
| C1 | 0.608 | 0.78 | C11 | 0.624 | 0.8 | 0.379 | 0.624 | C111 | 0.659 | 0.760 | 0.250 | 0.474 |
| | | | | | | | | C112 | 0.616 | 0.660 | 0.234 | 0.412 |
| | | | | | | | | C113 | 0.626 | 0.680 | 0.237 | 0.424 |
| | | | | | | | | C114 | 0.781 | 0.790 | 0.296 | 0.493 |
| | | | | | | | | C115 | 0.769 | 0.840 | 0.292 | 0.524 |
| | | | C12 | 0.709 | 0.79 | 0.431 | 0.616 | C121 | 0.616 | 0.660 | 0.266 | 0.407 |
| | | | | | | | | C122 | 0.540 | 0.610 | 0.233 | 0.376 |
| | | | | | | | | C123 | 0.799 | 0.750 | 0.344 | 0.462 |
| | | | | | | | | C124 | 0.616 | 0.640 | 0.266 | 0.394 |
| | | | C13 | 0.626 | 0.82 | 0.381 | 0.640 | C131 | 0.659 | 0.760 | 0.251 | 0.486 |
| | | | | | | | | C132 | 0.649 | 0.640 | 0.247 | 0.409 |
| | | | | | | | | C133 | 0.596 | 0.600 | 0.227 | 0.384 |
| | | | | | | | | C134 | 0.615 | 0.580 | 0.234 | 0.371 |
| | | | | | | | | C135 | 0.633 | 0.630 | 0.241 | 0.403 |
| | | | | | | | | C136 | 0.665 | 0.670 | 0.253 | 0.429 |
| | | | | | | | | C137 | 0.626 | 0.680 | 0.238 | 0.435 |
| | | | | | | | | C138 | 0.540 | 0.610 | 0.206 | 0.390 |
| | | | C14 | 0.709 | 0.69 | 0.431 | 0.538 | C141 | 0.799 | 0.750 | 0.344 | 0.404 |
| | | | | | | | | C142 | 0.781 | 0.790 | 0.337 | 0.425 |
| | | | | | | | | C143 | 0.769 | 0.870 | 0.331 | 0.468 |
| | | | C15 | 0.578 | 0.83 | 0.351 | 0.647 | C151 | 0.623 | 0.620 | 0.219 | 0.401 |
| | | | | | | | | C152 | 0.799 | 0.750 | 0.281 | 0.486 |
| | | | | | | | | C153 | 0.781 | 0.790 | 0.274 | 0.511 |
| | | | | | | | | C154 | 0.769 | 0.840 | 0.270 | 0.544 |
| C2 | 0.619 | 0.82 | C21 | 0.624 | 0.8 | 0.386 | 0.656 | C211 | 0.659 | 0.760 | 0.254 | 0.499 |
| | | | | | | | | C212 | 0.616 | 0.660 | 0.238 | 0.433 |
| | | | | | | | | C213 | 0.626 | 0.680 | 0.242 | 0.446 |
| | | | | | | | | C214 | 0.781 | 0.790 | 0.302 | 0.518 |
| | | | | | | | | C215 | 0.769 | 0.840 | 0.297 | 0.551 |
| | | | C22 | 0.709 | 0.79 | 0.439 | 0.648 | C221 | 0.616 | 0.660 | 0.270 | 0.428 |
| | | | | | | | | C222 | 0.540 | 0.610 | 0.237 | 0.395 |
| | | | | | | | | C223 | 0.799 | 0.750 | 0.351 | 0.486 |
| | | | | | | | | C224 | 0.616 | 0.640 | 0.270 | 0.415 |
| | | | C23 | 0.626 | 0.82 | 0.387 | 0.672 | C231 | 0.659 | 0.760 | 0.255 | 0.511 |
| | | | | | | | | C232 | 0.649 | 0.640 | 0.251 | 0.430 |
| | | | | | | | | C233 | 0.596 | 0.600 | 0.231 | 0.403 |
| | | | | | | | | C234 | 0.615 | 0.580 | 0.238 | 0.390 |
| | | | | | | | | C235 | 0.633 | 0.630 | 0.245 | 0.424 |
| | | | | | | | | C236 | 0.665 | 0.670 | 0.258 | 0.451 |
| | | | | | | | | C237 | 0.626 | 0.680 | 0.243 | 0.457 |
| | | | | | | | | C238 | 0.540 | 0.610 | 0.209 | 0.410 |
| | | | C24 | 0.61 | 0.81 | 0.483 | 0.648 | C241 | 0.781 | 0.790 | 0.378 | 0.512 |
| | | | C25 | 0.671 | 0.85 | 0.415 | 0.697 | C251 | 0.615 | 0.580 | 0.255 | 0.404 |
| | | | | | | | | C252 | 0.633 | 0.630 | 0.263 | 0.439 |
| | | | | | | | | C253 | 0.665 | 0.670 | 0.276 | 0.467 |

(Continued)

| The first level | The ratings of durability factors of the first level | | The second level | Local ratings of the second level | | The final ratings of the second level | | The level of the third level | The local ratings of the third level | | The final ratings of the third level | |
|---|---|---|---|---|---|---|---|---|---|---|---|---|
| | Version 1 | Version 2 | | Version 1 | Version 2 | Version 1 | Version 2 | | Version 1 | Version 2 | Version 1 | Version 2 |
| C3 | 0.595 | 0.87 | C31 | 0.709 | 0.79 | 0.422 | 0.687 | C311 | 0.616 | 0.660 | 0.260 | 0.454 |
| | | | | | | | | C312 | 0.540 | 0.610 | 0.228 | 0.419 |
| | | | | | | | | C313 | 0.799 | 0.750 | 0.337 | 0.515 |
| | | | | | | | | C314 | 0.616 | 0.640 | 0.260 | 0.440 |
| | | | C32 | 0.628 | 0.88 | 0.374 | 0.766 | C321 | 0.623 | 0.620 | 0.233 | 0.475 |
| | | | | | | | | C322 | 0.799 | 0.750 | 0.299 | 0.574 |
| | | | | | | | | C323 | 0.781 | 0.640 | 0.292 | 0.490 |
| | | | | | | | | C324 | 0.769 | 0.840 | 0.287 | 0.643 |
| | | | C33 | 0.61 | 0.81 | 0.363 | 0.705 | C331 | 0.781 | 0.790 | 0.283 | 0.557 |
| | | | C34 | 0.709 | 0.69 | 0.422 | 0.600 | C341 | 0.799 | 0.750 | 0.337 | 0.450 |
| | | | | | | | | C342 | 0.781 | 0.790 | 0.329 | 0.474 |
| | | | | | | | | C343 | 0.769 | 0.840 | 0.324 | 0.504 |
| | | | C35 | 0.548 | 0.83 | 0.326 | 0.722 | C351 | 0.623 | 0.620 | 0.203 | 0.448 |
| | | | | | | | | C352 | 0.799 | 0.750 | 0.261 | 0.542 |
| | | | | | | | | C353 | 0.781 | 0.790 | 0.255 | 0.570 |
| | | | | | | | | C354 | 0.769 | 0.840 | 0.251 | 0.607 |

**Table 30 Overall security durability.**

| | Version 1 | Version 2 |
|---|---|---|
| Security durability | 0.2852 | 0.4700 |

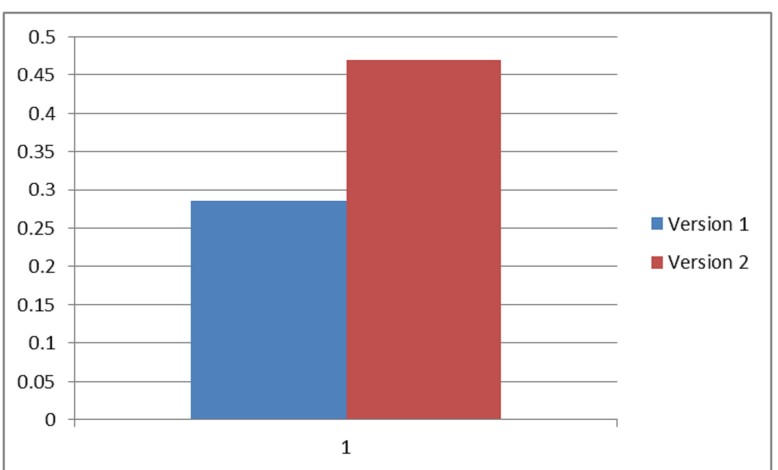

**Figure 6 Graphical representation of overall security durability.**

Table 30 and Fig. 6 are showing the values of security durability of BBAU software. Value of security durability for the old version (version 1) is 0.2852 and value of security durability for modified version (version 2) is 0.4700. Again, with the help of final weights,

| S. no. | Characteristics of level 1 | Version 1 | Version 2 |
|--------|---------------------------|-----------|-----------|
| | **Table 31** Security durability impact at level 1. | | |
| | The contribution of security durability at level 1 | | |
| 1 | Dependability | 0.1391 | 0.2187 |
| 2 | Trustworthiness | 0.0782 | 0.1246 |
| 3 | Human trust | 0.0679 | 0.1267 |

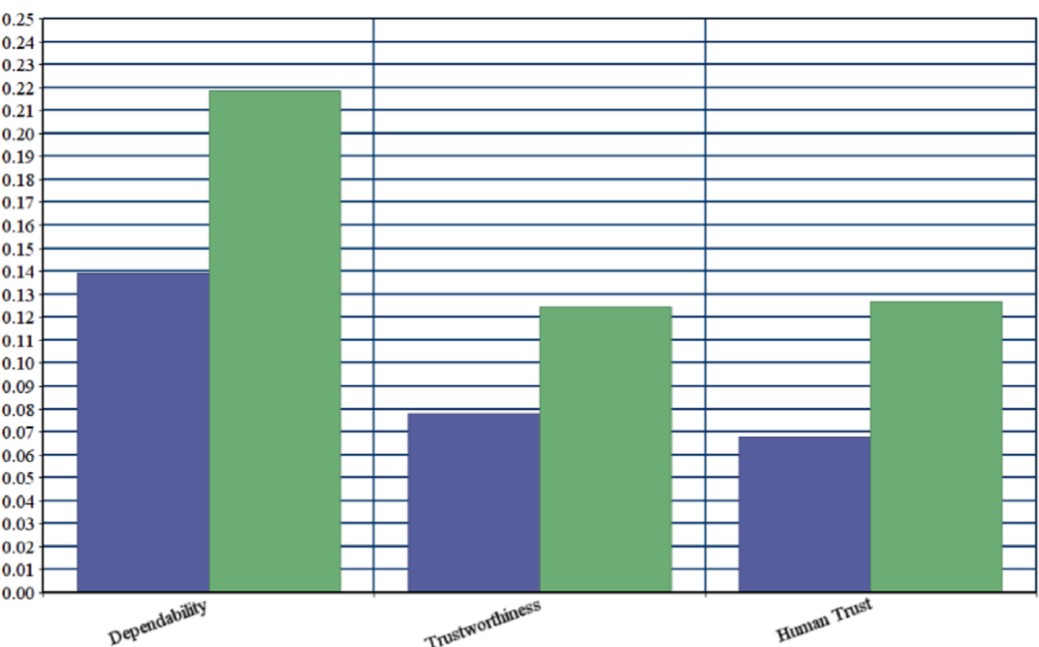

**Figure 7 Graphical representation of security durability impact at level 1.**

final ratings of both version, and Eq. (16), the impact of security durability at first level are calculated which is shown in Table 31.

Table 31 and Figure 7 are showing the values of security durability on first level attributes. Again, with the help of final weights, final ratings of both version and Eq. (16), the impact of security durability at the second level are calculated which is shown in Table 32.

Table 32 and Figure 8 enlist the values of security durability on second level attributes. Again, with the help of final weights, final ratings of both version and Eq. (16), the impact of security durability at third level are calculated which has been presented in Table 33.

Table 33 and Figure 9 are showing the values of security durability on third level attributes.

## Sensitivity analysis of the results

The technique used to determine how independent variable values will impact a particular dependent variable under a given set of assumptions is defined as sensitivity analysis. Sensitivity analysis also focuses on analyzing the effects of changes in key values of the

**Table 32 Security durability impact at level 2.**

| S. no. | Characteristics of level 2 | Version 1 | Version 2 |
|--------|---------------------------|-----------|-----------|
| | The contribution of security durability at level 2 | | |
| 1 | Reliability | 0.0584 | 0.0903 |
| 2 | Availability | 0.0237 | 0.0433 |
| 3 | Authentication | 0.0456 | 0.0931 |
| 4 | Maintainability | 0.0227 | 0.0403 |
| 5 | Confidentiality | 0.0696 | 0.0955 |
| 6 | Accountability | 0.0326 | 0.0502 |
| 7 | Consumer Integrity | 0.0108 | 0.0214 |
| 8 | Survivability | 0.0219 | 0.0360 |

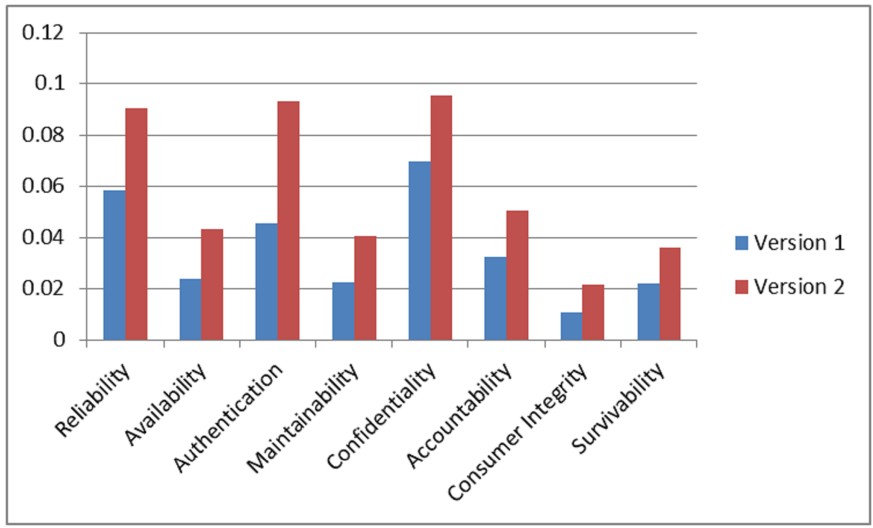

**Figure 8 Graphical representation of security durability impact at level 2.**

project and depends upon one or more input variables within the specific boundaries. Authors have taken the values of α and β as 0.5 and 0.5, respectively, during the defuzzification. The range of these two values ranges between 0 and 1, in such a way that a lesser value indicates greater uncertainty in decision making to preferences and risk tolerance of the participants. A total of 0.5 value for α and β is used to represent a balanced environment because the values of α and β are dependent on environmental uncertainties. This indicates that participants are neither extremely optimistic nor pessimistic about their judgments. These values will directly affect the weights of individual criteria, priority ranking and overall assessment of security durability.

If the participants involved in priority assessment have strong background knowledge of software security, the values of α and β can be readjusted to indicate confident judgments. Further, the sets of α and β values are 81 (9 × 9) including (0.1, 0.1), (0.1, 0.2), (0.2, 0.1), (0.1, 0.3), (0.3, 0.1), etc. The accuracy of Fuzzy AHP can be further improved

**Table 33  Security durability impact at level 3.**

The contribution of security durability at level 3

| S. no. | Characteristics of Level 3 | Version 1 | Version 2 |
| --- | --- | --- | --- |
| 1 | Software effectiveness evaluation | 0.0641 | 0.1014 |
| 2 | User satisfaction | 0.0344 | 0.0490 |
| 3 | Feasibility | 0.0239 | 0.0385 |
| 4 | Operational controls | 0.0758 | 0.1310 |
| 5 | Time-efficiency | 0.0136 | 0.0240 |
| 6 | Auditability | 0.0071 | 0.0137 |
| 7 | Psychological acceptability | 0.0094 | 0.0185 |
| 8 | Business continuity | 0.0167 | 0.0263 |
| 9 | Accessibility | 0.0043 | 0.0079 |
| 10 | Extensibility | 0.0118 | 0.0198 |
| 11 | Flexibility | 0.0101 | 0.0173 |
| 12 | Detectability | 0.0105 | 0.0167 |
| 13 | Scalability | 0.0012 | 0.0021 |
| 14 | Traceability | 0.0023 | 0.0039 |

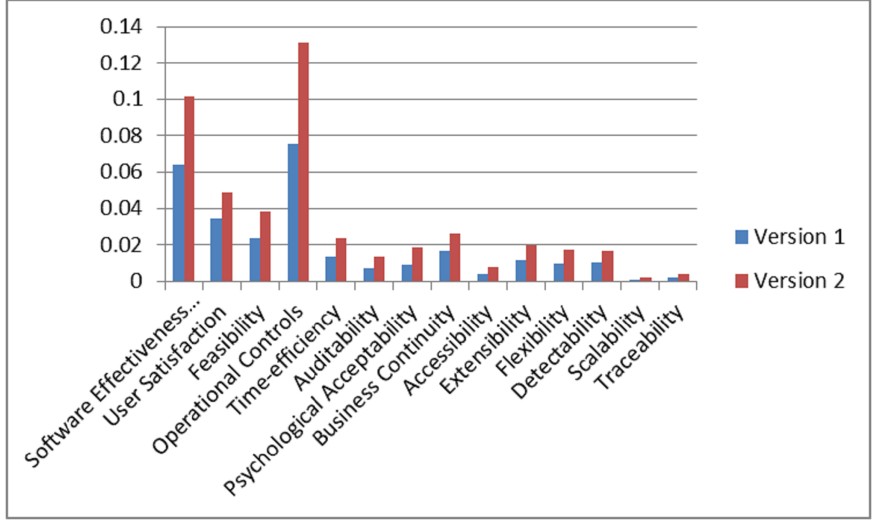

**Figure 9  Graphical representation of security durability impact at level 3.**

by investigating the impact of α and β values on the final results and analysis is needed in order to determine the values of α and β truthfully. That's why, to check the variations in the results, authors have used ten combinations of α and β values for version 1 and version 2 as experiment including E1 (0.1, 0.1), E2 (0.5, 0.1), E3 (0.5, 0.3), E4 (0.5, 0.7), E5 (0.5, 0.9), E6 (0.1, 0.5), E7 (0.3, 0.5), E8 (0.7, 0.5), E9 (0.9, 0.5), E10 (0.9, 0.9) with E0 (0.5, 0.5). Further, the value of α is constant for E2, E3, E4, E5, and value of β is in variation. While, the value of β is constant for E6, E7, E8, E9, and value of α is in variation. The results are shown in Table 34.

**Table 34 Sensitivity analysis due to $\alpha$ and $\beta$ values.**

**Sensitivity analysis**

| | E1 | | E2 | | E3 | | E4 | | E5 | | E0 | | E6 | | E7 | | E8 | | E9 | | E10 | |
|---|---|---|---|---|---|---|---|---|---|---|---|---|---|---|---|---|---|---|---|---|---|---|
| | Vers. 1 | Vers. 2 | Vers. 1 | Vers. 2 | Vers. 1 | Vers. 2 | Vers. 1 | Vers. 2 | Vers. 1 | Vers. 2 | Vers. 1 | Vers. 2 | Vers. 1 | Vers. 2 | Vers. 1 | Vers. 2 | Vers. 1 | Vers. 2 | Vers. 1 | Vers. 2 | Vers. 1 | Vers. 2 |
| Experiment number | E1 | | E2 | | E3 | | E4 | | E5 | | E0 | | E6 | | E7 | | E8 | | E9 | | E10 | |
| (Preferences of participants) $\alpha$ | 0.1 | | 0.5 | | 0.5 | | 0.5 | | 0.5 | | 0.5 | | 0.1 | | 0.3 | | 0.7 | | 0.9 | | 0.9 | |
| (Risk tolerance of participants) $\beta$ | 0.1 | | 0.1 | | 0.3 | | 0.7 | | 0.9 | | 0.5 | | 0.5 | | 0.5 | | 0.5 | | 0.5 | | 0.9 | |
| Security durability | 0.4642 | 0.6906 | 0.3687 | 0.5799 | 0.3263 | 0.5190 | 0.2465 | 0.4091 | 0.2110 | 0.3555 | 0.2852 | 0.4700 | 0.2910 | 0.4579 | 0.2878 | 0.4592 | 0.2789 | 0.4605 | 0.2751 | 0.4652 | 0.2427 | 0.4185 |

**Note:**
Vers. 1=Version 1; Vers. 2=Version 2.

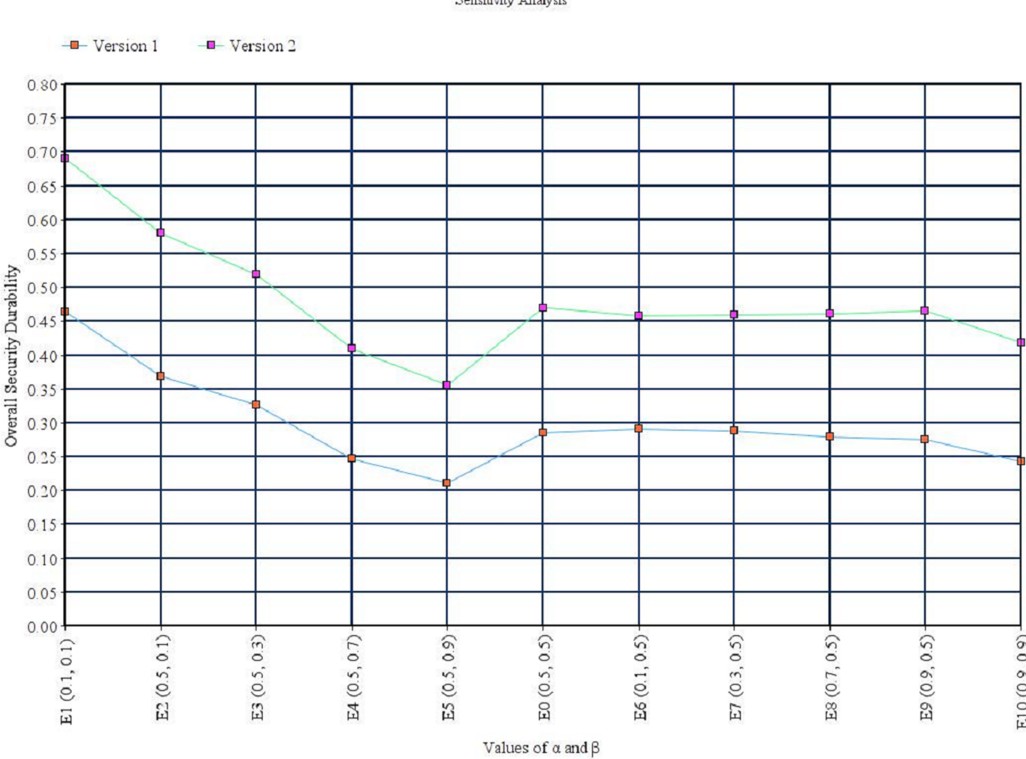

**Figure 10 Graphical representation of sensitivity analysis.**

Table 34 and Figure 10 show the variation in results due to $\alpha$ and $\beta$ values. Although, E0 (0.5, 0.5) gives the concentrated values of security durability including 0.2852, 0.4700 for version 1 and version 2, respectively. The results through the values of $\alpha$ and $\beta$ (as 0.5) indicate that a balanced environment about expert's judgments may give the best results. After going through the results of sensitivity analysis it has been determined that variation in the values of overall security durability is not negligible. Preferences of participants and risk tolerance of participants do have a considerable impact on the value of security durability.

## DISCUSSION

A series of tragedies and chaos caused by the insecure software proves that the duration of software security may become a grave matter of life and death at the time. Software industries are now focusing on longer security services of software as a major concern. Software security measurement and improvement have been one of the most talked about topics in organizations. In addition, identifying and addressing various security attributes during software development may reduce maintenance time and costs incurred. Security durability may be considered as one of the supporting attributes of security. Because durability strengthens the fact that longer security doesn't need maintenance for a specific duration. This decreases the cost and time invested in maintenance. Security durability assessment may intensely influence the security of the software.

The investigation of security durability parameters and their effect on security will ease to reveal the qualities and shortcomings of the security strength. The precise estimation of security durability remains a vital issue in light of the fact that there is supposedly no great comprehension of the idea of security durability. There is no unmistakable definition to "what perspectives are identified with security durability." Finding an appropriate method to measure security durability and the greater part of the angles identified with it is exceptionally troublesome. Hence, an examination of security durability assessment remains vital for security developers, programming engineers, and their clients. Durability applies a methodology that conveys robust, vibrant security to support, facilitate all business initiatives, including clouds, mobility, and improve security. The main advantages of security durability assessment are given below:

- Improved probability of lifetime of security software.
- Reduced cost of maintenance on security development life cycle.
- Reduced maintenance and repair costs of software security.
- Improved satisfaction of user's and market value of the product.
- Prioritized security durability attributes and guidelines may be helpful in designing secure as well as durable software.
- Field of security is still in its infancy and only quantitative assessment of security durability may facilitate the mechanism on predicting how long the software is secured.
- Since quantitative assessment techniques for security durability are not available, the security community primarily uses qualitative assessment techniques for security. The proposed study may help industry professionals in producing a quantitative estimation of security durability.

A consistent quantitative estimate of security durability is highly desirable for secure software during the development life cycle. The literature survey reveals that nothing significant, precise, and clear exists in this regard that can be used to quantify security durability in the early stage of development. Therefore, in absence of any framework or model for quantifying security durability, it is worthwhile to develop a methodology for security durability quantification. The main aim of this research is to gain an in-depth understanding of the durable security/security durability concept and the need to design durable as well as secure software.

Every coin has two sides. From the research point of view both surfaces hold imperative positions and are tenable. However, the positive appearance offers new dimensions to the proposed study while the negative portion highlights the deficiencies of work. After resolving the deficiencies of the intended work, the redesigned efforts ascertain innovative features of lessons. Despite having so many reasons favorable for the industrial adaptation of the approach, there are negative aspects also. Some are listed as follows:

- The approach is assessed with only 20 experts. The expert group may be larger for big datasets. Small group of experts may negotiate with the results.

- Due to unavailability of big industry data, the proposed framework is only validated with a small set of data which may further affect the overall results.
- The approach has used security metrics for improvement which has been derived from previous work. A specific security metric for security durability assessment can be developed.
- To provide more attention on security durability quantification area, only a set of security attributes and durability attributes have been chosen from the various security attributes and durability attributes, respectively. There can be more specific attributes of security durability and they may be integrated later for better results.

## CONCLUSION

The software security area of software engineering has been largely ignored since the birth of software. There may be several reasons for this. There was an era in which software security was an easy task and was achieved by applying only some passwords or installing some software. As the time passed, complex antivirus software has replaced easy-to-install software. The multiple connections making a policy of computer make it vulnerable to any virus and thus making it insecure for handling personal and sensitive information. Though there has been lot of work done in the field of software security to achieve maximum security in less time and cost, security also needs maintenance. The cost and time incurred on maintenance are increasing day by day. To reduce the maintenance time and cost and to improve the security life span of software, estimation of security durability will help in minimizing time and cost on the maintenance for a specific time period. On the successful completion of the study, the researchers found that early security durability estimation is highly desirable in the area of secure software development.

### Funding
Funding for this work was provided by the College of Computer and Information Sciences, Prince Sultan University. The funders had no role in study design, data collection and analysis, decision to publish, or preparation of the manuscript.

### Grant Disclosures
The following grant information was disclosed by the authors:
College of Computer and Information Sciences, Prince Sultan University.

### Competing Interests
The authors declare that they have no competing interests.

### Author Contributions
- Alka Agrawal conceived and designed the experiments, performed the experiments, analyzed the data, contributed reagents/materials/analysis tools, authored or reviewed drafts of the paper, approved the final draft.

- Mohammad Zarour prepared figures and/or tables, approved the final draft.
- Mamdouh Alenezi performed the computation work, approved the final draft.
- Rajeev Kumar conceived and designed the experiments, performed the experiments, contributed reagents/materials/analysis tools, prepared figures and/or tables, performed the computation work, approved the final draft.
- Raees Ahmad Khan analyzed the data, contributed reagents/materials/analysis tools, authored or reviewed drafts of the paper, approved the final draft.

## Data Availability

The raw data are available in the Supplemental Files.

## Supplemental Information

Supplemental information for this article can be found online at http://dx.doi.org/10.7717/peerj-cs.215#supplemental-information.

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
