# Peer review of "Security durability assessment through fuzzy analytic hierarchy process"

_PeerJ Computer Science, doi:10.7717/peerj-cs.215_

## Round 0.1 · original submission · Minor Revisions

The paper is suitable for publication but some minor revisions are required as per the comments below.

Reviewer 1 ·

Basic reporting

The paper has made several contributions toward the establishment of security durability during software development in industry. The manuscript introduces the security durability and assessment. The quality of the article is good, the content is novel and the structure is complete.
The work presented should be more precise and related to work done in the paper. Too many general stiffs discussed should be avoided. It is suggested to carefully check the format.
Recommended for inclusion.

Experimental design

Entrance exam software of BBA University has been taken as an experiment. But the attributes of design hasn't been provided. Although, it is made clear that only security attributes are taken. But I think more attributes of design and security can be merged together to form the hierarchy.

Validity of the findings

Fuzzy MCDM methods have taken to assess the security durability and use the values of alpha and beta as 0.5. The researcher is given the sensitivity analysis of the results with respect to value of alpha and beta. More methods of MCDM can be used for better validation of the results.

Additional comments

Please add more security analysis on the proposed system.

Reviewer 2 ·

Basic reporting

Please revise the format of all tables.

Experimental design

This paper provides original results of tha application of a multicriteria methods already largely used. In terms of relevance it was not clear to if the merit of this paper. So, please better explain the relevance of its paper as it is a crucial aspect to approuve its publication.

Validity of the findings

In addition below I provide other comments to be addressed:
Abstract
Methods: description of methods in abstract is very generalist, please revise this part focused on methods and procedures used.
Results: description of results in abstract provide no usefull information, please revise it.
Material and methods
Line 206 to 258. This part is not material and methods, it is literature review, please move it to the correct section.
Line 259: AHP is one of the most largely used multicriteria methods and it presents several disadvantages over other methods like compensation and rank reversal. So, please justify why AHP was adopted instead of other methods that does not present its disadvantages.
Line 468 to 490. This part is not material and methods, it is literature review, please move it to the correct section.
Line 750 to 865. This part was very generalist and detached from the rest of this section, please condense the information provided here.

·

Basic reporting

The text is well structured but the main objective of the article was not shown in the abstract clearly. It is better the introduction section rewrite.
You need to present the text structure at the end of the Introduction.

Experimental design

No comment

Validity of the findings

No comment

Additional comments

I’d like to suggest that the authors get editing help from someone with full professional proficiency in English.

---

## Round 0.2 · accepted · Accept

All the minor revisions are made. I think the paper is now ready for publication.